# Off-Policy Evaluation with Deficient Support Using Side Information

**Nicolò Felicioni**
Politecnico di Milano
nicolo.felicioni@polimi.it

**Maurizio Ferrari Dacrema**
Politecnico di Milano
maurizio.ferrari@polimi.it

**Marcello Restelli**
Politecnico di Milano
marcello.restelli@polimi.it

**Paolo Cremonesi**
Politecnico di Milano
paolo.cremonesi@polimi.it

## Abstract

The Off-Policy Evaluation (OPE) problem consists in evaluating the performance of new policies from the data collected by another one. OPE is crucial when evaluating a new policy online is too expensive or risky. Many of the state-of-the-art OPE estimators are based on the Inverse Propensity Scoring (IPS) technique, which provides an unbiased estimator when the full support assumption holds, i.e., when the logging policy assigns a non-zero probability to each action. However, there are several scenarios where this assumption does not hold in practice, i.e., there is deficient support, and the IPS estimator is biased in the general case. In this paper, we consider two alternative estimators for the deficient support OPE problem. We first show how to adapt an estimator that was originally proposed for a different domain to the deficient support setting. Then, we propose another estimator, which is a novel contribution of this paper. These estimators exploit additional information about the actions, which we call side information, in order to make reliable estimates on the unsupported actions. Under alternative assumptions that do not require full support, we show that the considered estimators are unbiased. We also provide a theoretical analysis of the concentration when relaxing all the assumptions. Finally, we provide an experimental evaluation showing how the considered estimators are better suited for the deficient support setting compared to the baselines.

## 1 Introduction

Many real-world decision-making problems can be viewed through the lens of the contextual bandit framework. Some prominent examples are medical treatments [19], recommendation systems [15, 18, 31, 48], search engines [32], ad-placement systems [9], and many others. In any of these problems, we have a decision-maker who repeatedly observes a context (e.g., a profile of a patient), samples an action according to its policy (e.g., provides a medical treatment), and collects a reward (e.g., +1 if the patient survives). Also, in many of those applications, we have access to additional information about the actions, which we call *side information*. For example, in movie recommendation, we may know the director and the actors of each movie [12]; in a clinical trial, we may have access to a set of characteristics of each drug [10].

While deploying and evaluating a new policy on a real system may be prohibitively expensive (if not unfeasible), logged data of contextual bandit problems is relatively cheap to obtain. Hence, it is desirable to use them to evaluate new policies, without the need to collect new data. This problem is called *Off-Policy Evaluation* (OPE). In this case, we have data collected by a given logging policy, and

36th Conference on Neural Information Processing Systems (NeurIPS 2022).

we want to evaluate a different policy, called evaluation policy. One way to think about this problem is that we are trying to answer the following counterfactual question: "*What would have happened if the evaluation policy was deployed instead of the logging one?*". One of the most widely employed estimators for OPE is the *Inverse Propensity Score* (IPS), along with its variants [43, 48, 56, 61]. The IPS technique tries to de-bias the collected rewards by accounting for the propensity of the logging policy. One of the reasons for the widespread adoption of the IPS estimator is that it is unbiased under the *full support* assumption. The full support assumption states that no action that is possible under the evaluation policy has zero probability under the logging policy. Unfortunately, we argue that this assumption is unrealistic in many real-world systems. A practical example is a recommendation system that adopts a user-based pre-processing of the items to recommend, discarding some items for a given user to improve scalability. In this way, some actions (i.e., items) will never be proposed to a given context (i.e., a user), violating the full support assumption. Another example is the *new action* problem, namely, when the action space is expanded after the logging phase. This problem is typical of many real-world use cases, such as recommendation systems [4, 29, 51, 52], drug interaction evaluation [33], clinical trials [40], etc. In general, whenever the full support assumption is not valid, we say that we have an OPE problem with *deficient support*. If this is the case, the IPS estimator can be highly biased [47]. A possible mitigation for this issue is using a model-based approach, which means training a regression model that aims to approximate the reward function and extrapolate the rewards for the unsupported actions [5, 6, 13, 35, 47]. The drawback of this method is that model misspecification can lead to a high bias [13, 35, 50].

In this paper, we focus on estimators without a regression model for deficient support that exploit side information about the actions. After introducing the necessary background (Section 2), we consider two alternative estimators for Off-Policy Evaluation with deficient support in Section 3. While relaxing the full support assumption, we offer alternative assumptions that hold even with deficient support. In Section 3.1, we present the *PseudoInverse* estimator. This estimator was introduced by Swaminathan et al. [63] in the context of slate recommendation. We show how this estimator can be adapted to the presence of side information, and we show that it is unbiased under two alternative assumptions: *full support on side information* and *linearity*. Full support on side information is a milder assumption than full support: it requires the logging policy to be able to select each feature. Linearity, instead, requires that the rewards are linear combinations of the action features. This assumption is already used in other contexts, such as recommendation systems [27] and online linear bandits [2, 11]. In Section 3.2, we propose a novel estimator, which we call *Similarity* estimator. It is based on the assumption that expected rewards of similar actions are similar. Under this assumption, we prove its unbiasedness, even in the presence of deficient support. In Section 3.3, we relax all the assumptions and derive finite-sample concentration inequalities of the considered estimators for deficient support and the traditional IPS to better understand the theoretical guarantees of each of them. In Section 4, we provide an experimental evaluation, showing that the considered estimators consistently outperform traditional approaches (such as IPS and regression-based estimators) on a real-world dataset with deficient support from the recommendation systems domain. Finally, in Section 5 we review related work, and in Section 6 we draw conclusions.

**Societal Impacts** Improving OPE may be beneficial to society as it allows decision-makers to assess potentially dangerous policies without testing them on a real system. This is even more relevant in the presence of deficient support, when there are context-action pairs that are never observed in the data.

## 2  Background and Preliminaries

First of all, we introduce the *Off-Policy Evaluation* (OPE) problem in the contextual bandit setting with side information on the actions, and then, we describe the deficient support problem.

**Off-Policy Evaluation** Let $x$ be a context sampled from a prior (unknown) context distribution $p$ over the context space $X$, i.e., $x \sim p(\cdot)$. An action $a$ is sampled from a probability $\pi$ (called *policy*) over the action space $A$, conditioned on the observed context $x$, i.e., $a \sim \pi(\cdot|x)$. After this choice, a reward $r$ is observed from the reward distribution conditioned on the observed context and chosen action, i.e., $r \sim p(\cdot|x, a)$. Without loss of generality, we consider as reward space $[0, 1]$. To evaluate a policy, we use the *policy value* $R(\pi) := \mathbb{E}_{x \sim p(\cdot)} \mathbb{E}_{a \sim \pi(\cdot|x)} \mathbb{E}_{r \sim p(\cdot|x,a)}[r]$. It is useful to define the expected reward given a context and an action: $\delta(x, a) := \mathbb{E}_{r \sim p(\cdot|x,a)}[r]$. With this function, we

can simplify the policy value formulation: $R(\pi) = \mathbb{E}_{x \sim p(\cdot)} \mathbb{E}_{a \sim \pi(\cdot|x)}[\delta(x, a)]$. In this paper, we focus on the contextual bandit problem with side information about the actions. Therefore, we have access to action features, which are represented as vectors: $f(a) \in \mathbb{R}^F$ for any $a \in A$. We focus on the *Off-Policy Evaluation* (OPE) problem. We call the logging policy $\pi_0$. Therefore, the collected dataset will be in the following form: $\mathcal{D} := \{x_i, a_i, r_i, \pi_0(a_i|x_i)\}_{i=1}^n$, where, for each $i$, $x_i \sim p(\cdot)$, $a_i \sim \pi_0(\cdot|x_i)$, and $r_i \sim p(\cdot|x_i, a_i)$. Given this dataset, we would like to evaluate another policy by estimating its value function. The new policy is called the *evaluation policy* $\pi_e$. Thus, we want to find a plausible estimator $\hat{R}(\pi_e)$ using the given dataset ($\hat{R}(\pi_e) = \hat{R}(\pi_e; \mathcal{D})$) such that $\hat{R}(\pi_e) \approx R(\pi_e)$. One of the most used estimators for OPE is the *Inverse Propensity Score* (IPS), along with its variants. The IPS estimator is defined as $\hat{R}_{\text{IPS}}(\pi_e) = \frac{1}{n} \sum_{i=1}^n \frac{\pi_e(a_i|x_i)}{\pi_0(a_i|x_i)} r_i$. A fundamental property of such estimator is that it is an *unbiased* estimator of the evaluation policy value if the *full support assumption* holds [31]. Let us define $supp(f) := \{z \mid f(z) > 0\}$, $A_0(x) := supp(\pi_0(\cdot|x))$, and $A_e(x) := supp(\pi_e(\cdot|x))$, for any $x$. The full support assumption is stated in the following.

**Assumption 1** (Full Support). *The off-policy evaluation problem satisfies the full support assumption if $A_e(x) \subseteq A_0(x)$ with probability one for $x \sim p(\cdot)$.*

**Deficient Support**  Whenever the full support assumption is not valid, we say that we have an OPE problem with *deficient support* [47]. If this is the case, the IPS estimator is no longer unbiased. Let us define the set of unsupported actions for context $x$ under $\pi_0$ as $\text{Un}(x, \pi_e, \pi_0) := A_e(x) \setminus A_0(x)$. The bias of the IPS depends on this set, as illustrated by the following proposition.

**Proposition 1** ([47], Proposition 1). *In an off-policy evaluation problem, the bias of $\hat{R}_{IPS}(\pi_e)$ is equal to the negative expected reward on the unsupported action set:*

$$bias(\hat{R}_{IPS}(\pi_e)) = \mathbb{E}_{x \sim p(\cdot)} \left[ - \sum_{a \in Un(x, \pi_e, \pi_0)} \pi_e(a|x)\delta(x, a) \right]$$

This finding is intuitive: if the unsupported set is empty (i.e., the full support assumption is valid), the IPS estimator has no bias. Otherwise, if there are unsupported actions, the IPS estimator has a bias caused by the blind areas in the dataset, where there are actions for which we never observe the reward.

## 3 Estimators for Off-Policy Evaluation with Side Information

In this section, we first introduce two estimators for a better off-policy evaluation when we have deficient support and side information about the actions, and then we provide finite-sample error bounds. The proofs of the mathematical statements can be found in Appendix A.

### 3.1 PseudoInverse Estimator

The *PseudoInverse* (PI) estimator was initially proposed by Swaminathan et al. [63] for the problem of off-policy evaluation for *slate* recommendation [23]. In the following, we will show that we can use an analogous estimator for the OPE problem with side information.

We begin by describing two assumptions, which are relatively mild conditions and can also hold in the presence of deficient support.

**Assumption 2** (Full Support on Side Information). *The off-policy evaluation problem satisfies the full support on side information if, whenever $\pi_e(a|x) > 0$, this implies that, for every feature of $a$ (i.e., $\forall j \in \{j|f(a)_j > 0\}$) there exists an action $a' \in A_0(x)$ that has the same feature (i.e., $f(a')_j > 0$) with probability one over $x \sim p(\cdot)$.*

We notice that this assumption can also be satisfied in the presence of deficient support, as long as the logging policy can propose each feature. Now we make an assumption on the reward structure.

**Assumption 3** (Linearity). *For each context $x \in X$ there exists an (unknown) intrinsic reward vector $\phi_x \in \mathbb{R}^F$ such that $\delta(x, a) = f(a)^T \phi_x$.*

Intuitively, this assumption says that each action feature contributes linearly to the final reward. This assumption found applications in different fields: in recommendation systems, it constitutes the basis

of the Matrix Factorization algorithm [27], where we have fixed the latent item factors to be the feature vectors; in bandits literature, it corresponds to the linear bandit formulation [2, 11].

Hence, we can view this problem as a regression problem, where we interpret the $\phi_x$ vector as the weight vector $w$ to learn and we want to minimize the following error (for each $x$): $\mathbb{E}_{a \sim \pi_0(\cdot|x)} \mathbb{E}_{r \sim p(\cdot|x,a)}[(f(a)^T w - r)^2]$. The PseudoInverse estimator derives from the minimization of this error, and it is unbiased under the two previously described assumptions.

**Theorem 1.** *Consider the off-policy evaluation problem where Assumptions 2 and 3 hold. Then, we can define an unbiased estimator $\hat{R}_{PI}(\pi_e)$ of the expected reward of the evaluation policy as:*

$$\hat{R}_{PI}(\pi_e) := \frac{1}{n} \sum_{i=1}^{n} r_i \cdot \mathbf{q}_{\pi_e,x_i}^T \mathbf{\Gamma}_{\pi_0,x_i}^\dagger f(a_i),$$

*where $\mathbf{\Gamma}_{\pi_0,x} := \mathbb{E}_{a \sim \pi_0(\cdot|x)}[f(a)f(a)^T] \in \mathbb{R}^{F \times F}$, $\mathbf{q}_{\pi_e,x} := \mathbb{E}_{a \sim \pi_e(\cdot|x)}[f(a)] \in \mathbb{R}^F$, and $M^\dagger$ is the Moore-Penrose pseudoinverse of a generic matrix $M$.*

For the details of the derivation, we refer to [63]. Inspired by [61], we can also use a *Self-Normalized* version of the estimator, which is a biased but consistent estimator with a reduced variance:

$$\hat{R}_{SN-PI}(\pi_e) := \frac{1}{\sum_{i=1}^{n} \mathbf{q}_{\pi_e,x_i}^T \mathbf{\Gamma}_{\pi_0,x_i}^\dagger f(a_i)} \sum_{i=1}^{n} r_i \cdot \mathbf{q}_{\pi_e,x_i}^T \mathbf{\Gamma}_{\pi_0,x_i}^\dagger f(a_i).$$

### 3.2 Similarity Estimator

In some cases, the calculation of the pseudoinverse of a matrix can be computationally expensive. Thus, in the following, we propose an estimator based on computationally cheaper operations, which stems from a different assumption. Such assumption comes from the recommendation systems research literature [46]. Two of the most common approaches in recommendation systems are *collaborative filtering* [45] and *content-based filtering* [36] methods. Both of these methods are based on the assumption that the reward of an item (i.e., an action in the contextual bandit setting) given a user (i.e., a context) can be approximated by a weighted sum of rewards observed from different items given the same user. In our setting, we can define this assumption as follows:

**Assumption 4.** *The off-policy evaluation problem satisfies the similarity assumption if, given any context $x \in X$, the following condition holds:*

$$\exists w_x : A_e(x) \times A_0(x) \to \mathbb{R} \text{ such that } \forall a \in A_e(x), \ \delta(x,a) = \sum_{a' \in A_0(x)} w_x(a,a')\delta(x,a')$$

$$\text{and} \sum_{a' \in A_0(x)} w_x(a,a') = 1.$$

This indicates that we can approximate the reward for any action with a weighted sum of rewards observed from the actions supported by the logging policy. The weighting function should sum up to one over the supported actions, and intuitively it should indicate how much two actions are perceived as similar for a given context.

The aforementioned assumption is also strictly tied with the *Lipschitz assumption*, which is widely exploited in the research field of online bandits with side information [25, 26, 41, 54, 55]. The connection between the two assumptions is exhibited by the Proposition 3, in Appendix A.

By exploiting the similarity assumption, we can get an unbiased estimator of the expected reward of the evaluation policy even with deficient support. We explain the procedure in the following.

**Theorem 2.** *Consider the off-policy evaluation problem where the similarity assumption holds. Then, we can define an unbiased estimator $\hat{R}_S(\pi_e)$ of the expected reward of the evaluation policy as:*

$$\hat{R}_S(\pi_e) := \frac{1}{n} \sum_{i=1}^{n} \frac{\bar{\pi}(a_i|x_i)}{\pi_0(a_i|x_i)} r_i,$$

*where $\bar{\pi}(a_i|x_i) := \mathbb{E}_{a \sim \pi_e(\cdot|x_i)}[w_{x_i}(a,a_i)]$.*

We notice that the meaning of the estimator is somehow intuitive. Each observed reward $r_i$ is divided by the propensity of the logging policy to eliminate the bias introduced by the data collection procedure (as in IPS). At the same time, it is amplified by how much, on average, the evaluation policy proposes an action similar to $a_i$ for the context $x_i$. Given that its structure is comparable to the one of the IPS, the Self-Normalized version of the Similarity estimator is a consistent estimator under the same assumptions of Theorem 2 [61]:

$$\hat{R}_{SN-S}(\pi_e) := \frac{1}{\sum_{i=1}^{n} \frac{\bar{\pi}(a_i|x_i)}{\pi_0(a_i|x_i)}} \sum_{i=1}^{n} \frac{\bar{\pi}(a_i|x_i)}{\pi_0(a_i|x_i)} r_i.$$

**Comparison with Regression-based Estimators**   While this estimator does not require a regression model, we show that, from a theoretical point of view, the similarity estimator has some analogies with a regression-based estimator. Within this analysis, we will also see the essential differences between the two types of estimators. First, we define the simplest and most used type of regression-based estimator, which we call *Direct Method* (DM) [5]:

$$\hat{R}_{DM}(\pi_e) = \frac{1}{n} \sum_{i=1}^{n} \sum_{a \in A_e(x_i)} \pi_e(a|x_i)\hat{\delta}(x_i, a),$$

where $\hat{\delta}$ is a regression model. The theoretical analysis will focus on the expected values of the estimators. To lighten the notation, we will omit the distributions of the expected value whenever the expectation is with respect to the data distribution. The expected value of DM is:

$$\mathbb{E}[\hat{R}_{DM}(\pi_e)] = \mathbb{E}_{x \sim p(\cdot)} \mathbb{E}_{a \sim \pi_e(\cdot|x)} \left[ \hat{\delta}(x, a) \right].$$

Now, let us analyze the expected value of the similarity estimator:

$$\mathbb{E}[\hat{R}_S(\pi_e)] = \mathbb{E}_{x \sim p(\cdot)} \mathbb{E}_{a' \sim \pi_0(\cdot|x)} \left[ \frac{\bar{\pi}(a'|x)}{\pi_0(a'|x)} \delta(x, a') \right]$$

$$= \mathbb{E}_{x \sim p(\cdot)} \left[ \sum_{a' \in A_0(x)} \bar{\pi}(a'|x)\delta(x, a') \right]$$

$$= \mathbb{E}_{x \sim p(\cdot)} \left[ \sum_{a' \in A_0(x)} \sum_{a \in A_e(x)} \pi_e(a|x)w_x(a, a')\delta(x, a') \right] \quad \text{(from the definition of } \bar{\pi})$$

$$= \mathbb{E}_{x \sim p(\cdot)} \mathbb{E}_{a \sim \pi_e(\cdot|x)} \left[ \sum_{a' \in A_0(x)} w_x(a, a')\delta(x, a') \right].$$

Therefore, we notice that the similarity estimator is analogous (in expectation) to a regression-based estimator where the regression model is $\hat{\delta}(x, a) = \sum_{a' \in A_0(x)} w_x(a, a')\delta(x, a')$. What the similarity estimator is doing is trying to overcome support deficiency by exploiting the expected reward on supported actions and the information on the similarity between supported and unsupported actions.

However, it would be erroneous to think that, because of this analogy, it suffices to create a regression-based estimator with the regression model $\hat{\delta}(x, a) = \sum_{a' \in A_0(x)} w_x(a, a')\delta(x, a')$. The problem is that, in general, it is not possible to directly estimate all the possible expected rewards $\delta(x, a')$ for any $a' \in A_0(x)$ from data. Indeed, the fact that $a' \in A_0(x)$ does not imply that the logging policy chose action $a'$, but only that the probability of selecting $a'$ was positive. This means that, in general, we may have never seen a pair $x, a'$ in the dataset, and consequently, we may have never observed any $r \sim p(\cdot|x, a')$. Therefore, the only way to exploit the similarity assumption is via the proposed Similarity estimator.

### 3.3   Concentration Analysis

Up until now, we have shown that the previous estimators are unbiased under some assumptions. In the following, we will analyze the concentration behavior of the PseudoInverse, Similarity, and traditional

IPS estimators. We will also relax every assumption and investigate the bias-variance trade-off that arises. To do so, we provide finite-sample error bounds. In order to lighten the notation, we will drop the distributions of the expected value whenever the expectation is for $(x, a, r) \sim p(\cdot)\pi_0(\cdot|x)p(\cdot|x, a)$. We will refer to the absolute value of the bias of an estimator as $\epsilon := |\mathbb{E}[\hat{R}(\pi_e)] - R(\pi_e)|$. We introduce two different functions to measure the difference between the conditional distributions induced by two policies. The first one is a premetric called *Support Divergence*, introduced by Sachdeva et al. [47]:

$$d^{supp}(\pi_e||\pi_0) := \mathbb{E}_{x \sim p(\cdot)}\left[\sum_{a \in \text{Un}(x, \pi_e, \pi_0)} \pi_e(a|x)\right].$$

The second one is the *Exponentiated Rényi Divergence* [37, 44]:

$$d_2(\pi_e||\pi_0) := \mathbb{E}_{\substack{x \sim p(\cdot) \\ a \sim \pi_0(\cdot|x)}}\left[\left(\frac{\pi_e(a|x)}{\pi_0(a|x)}\right)^2\right].$$

Now, we are ready to introduce finite-sample concentration inequalities.

**Proposition 2.** *Let us consider a generic off-policy evaluation problem with side information. Let $\hat{R}_{IPS}$ be the IPS estimator, $\hat{R}_S$ the similarity estimator with weighting function $w_x$, $\hat{R}_{PI}$ the pseudo-inverse estimator. Then, for any $\gamma \in (0, 1)$, the following inequalities hold with probability at least $1 - \gamma$:*

$$\left|\hat{R}_{IPS}(\pi_e) - R(\pi_e)\right| \leq d^{supp}(\pi_e||\pi_0) + \sqrt{\frac{d_2(\pi_e||\pi_0)}{n\gamma}},$$

$$\left|\hat{R}_S(\pi_e) - R(\pi_e)\right| \leq \epsilon^S + \sqrt{\frac{d_2(\bar{\pi}||\pi_0)}{n\gamma}},$$

$$\left|\hat{R}_{PI}(\pi_e) - R(\pi_e)\right| \leq \epsilon^{PI} + \sqrt{\frac{2\sigma_{PI}^2 \ln(2/\gamma)}{n}} + \frac{2(\rho_{PI} + 1)\ln(2/\gamma)}{3n},$$

*where* $\bar{\pi}(a|x) := \mathbb{E}_{a' \sim \pi_e(\cdot|x)}[w_x(a', a)]$, $\sigma_{PI}^2 := \mathbb{E}_{x \sim p(\cdot)}[\mathbf{q}_{\pi_e,x}^T \mathbf{\Gamma}_{\pi_0,x}^\dagger \mathbf{q}_{\pi_e,x}]$, $\rho_{PI} := \sup_x \sup_a |\mathbf{q}_{\pi_e,x}^T \mathbf{\Gamma}_{\pi_0,x}^\dagger \mathbf{f}(a)|$, *and* $\mathbf{q}_{\pi_e,x}, \mathbf{\Gamma}_{\pi_0,x}$ *are defined as in Theorem 1.*

The error bounds for both the IPS and the Similarity estimators are derived following the main idea by Metelli et al. [37]. In particular, they provide a bound for the standard IPS estimator based on a Chebyshev-like inequality using the Rényi divergence [44]. The bound for the PI estimator derives from Theorem 1 by Swaminathan et al. [63] after having taken the bias into account.

We can notice how all the bounds exhibit a similar pattern. Each inequality has a component that depends on the approximation error of the estimator ($d^{supp}(\pi_e||\pi_0), \epsilon^S, \epsilon^{PI}$, respectively). In particular cases, the approximation error of such estimators can be zero ($d^{supp}(\pi_e||\pi_0) = 0$ when there is full support; $\epsilon^S = 0$ when the similarity assumption holds; $\epsilon^{PI} = 0$ when there is full support on side information and the linearity assumption holds). Still, in general, we have to consider the error resulting from the possible violation of the assumptions. Furthermore, we can notice that this type of error does not decrease with the sample size $n$. Therefore, if present, it is an irreducible error.

On the other hand, all the bounds have a remaining component that decreases with the sample size and depends on the difference between the evaluation and logging policies. In particular, the first two inequalities display a *polynomial* concentration (the dependence on $n$ and $\gamma$ is $\mathcal{O}(\sqrt{1/n\gamma})$). This bound has been proven to be tight in [39] for the IPS estimator. For the third inequality, we have a better behavior due to the *exponential* concentration (when the variance $\sigma_{PI}^2$ is small the dependence on $n$ and $\gamma$ is $\mathcal{O}(\sqrt{\ln(1/\gamma)/n})$).

Finally, we notice that the first two inequalities are very similar since the IPS estimator and the Similarity one have a comparable structure, but the former depends on $d_2(\pi_e||\pi_0)$, while the latter on $d_2(\bar{\pi}||\pi_0)$. This indicates how the Similarity estimator has an additional degree of freedom for regulating the bias-variance trade-off. Indeed, by inducing a complex but accurate $\bar{\pi}$, it may decrease the bias factor $\epsilon^S$, while increasing $d_2(\bar{\pi}||\pi_0)$. On the other hand, with a simple model of $\bar{\pi}$, it may reduce the $d_2(\bar{\pi}||\pi_0)$ factor, but it may incur a high bias.

# 4 Experiments

In this section, we empirically evaluate the considered estimators in an OPE problem with real logged bandit feedback. In particular, we will show how the estimators that exploit side information are competitive in practice with traditional techniques, especially in the presence of deficient support. The code used for the experiments can be found at
`https://github.com/recsyspolimi/neurips-2022-ope-side-info`.

## 4.1 Setup

We mainly make use of two Python packages: *Open Bandit Pipeline* [49] and *PyIEOE* [50]. These two packages are open source and freely available[1],[2]. Open Bandit Pipeline is a library with many state-of-the-art Off-Policy estimators already implemented, while PyIEOE is a collection of scripts for evaluating and comparing OPE estimators.

The dataset that we use is the *Open Bandit Dataset* (OBD), released with Open Bandit Pipeline. OBD contains logged bandit feedback from a real-world application (a large-scale fashion e-commerce platform). There are three campaigns available, namely "ALL", "Men", and "Women". We select the "ALL" campaign. The dimension of the context vector $x$ is 20, and the number of actions $|A| = 80$. Importantly, this dataset contains side information regarding the actions (inside the dataset, called *action context*). The action context has 4 features for each action, among which 3 categorical and one real. We discard the last one and apply one-hot encoding on the first 3. In the end, we get a binary feature vector $f(a)$ for each $a$ with dimension 40.

## 4.2 Deficient Support Evaluation

OBD consists of two separate datasets collected with two different policies: one with a uniform random policy $\mathcal{D}_r = \{x_i, a_i, r_i, \pi_r(a_i|x_i) = \frac{1}{|A|}\}_{i=1}^{n_r}$, and the other with a Bernoulli Thompson Sampling (BTS) policy $\mathcal{D}_{\text{bts}} = \{x_i, a_i, r_i, \pi_{\text{bts}}(a_i|x_i)\}_{i=1}^{n_{\text{bts}}}$. Since both policies satisfy the full support assumption, we have to pre-process the data in order to simulate a deficient support scenario.

Specifically, we select the random policy as the logging one ($\pi_0 = \pi_r$), and the BTS as the evaluation policy ($\pi_e = \pi_{\text{bts}}$). Since we have a dataset collected with the BTS, we can easily compute its policy value by on-policy estimation, which we call $R_{\text{on}}(\pi_e) := \frac{1}{n_{\text{bts}}} \sum_{r \in \mathcal{D}_{\text{bts}}} r$. We select a set of random seeds, $\mathcal{S}$. We select a deficient support rate $p$. This rate represents the percentage of the actions that should have deficient support for each context under the logging policy.

For each random seed $s \in \mathcal{S}$, our pre-processing algorithm proceeds as follows: (i) Select a random sub-sample of $n^* = 100,000$ rows from the logging dataset $\mathcal{D}_r$; (ii) Select $p$ actions randomly for each context, obtaining $\text{Un}(x, \pi_e, \pi_0)$; (iii) For each context $x$, discard the data points $(x, a, r, \pi_0(a|r))$ where $a \in \text{Un}(x, \pi_e, \pi_0)$; (iv) Modify the logged propensity for each data point: since we know that the logging policy was a uniform random, we need to set $\pi_0(a|x) = 1/|A|(100\% - p)$; (v) Now, we can use the resulting dataset to get the estimate $\hat{R}(\pi_e)$ and get the squared error w.r.t. $R_{\text{on}}(\pi_e)$. This pre-processing protocol is summarized in Algorithm 1, presented in Appendix B.

## 4.3 Hyperparameter Selection

The proposed Similarity estimator has a weighting function to be set. In the following experimental evaluation, we present two possible variants. The crucial aspect is that they both exploit side information of the actions so that we have a weighting for unsupported actions. Other weighting functions can be proposed, but we leave this analysis as future work.

The first proposal is to have a weighting function proportional to the *cosine* similarity among side information: $w_x(a, a') \propto s_{\cos}(a, a') := f(a)^T f(a')/\|f(a)\|_2 \|f(a')\|_2$. The cosine similarity is typical in fields like recommendation systems [16] and information retrieval [53]. We scale the weighting function such that $\sum_{a' \in A_0(x)} w_x(a, a') = 1$ for any $a \in A_e(x)$ (in accordance with

---

[1]Open Bandit Pipeline available with Apache License 2.0: `https://github.com/st-tech/zr-obp`

[2]PyIEOE available with MIT License: `https://github.com/sony/pyIEOE`

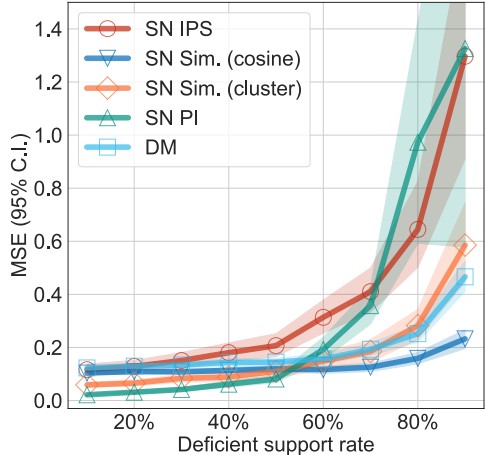
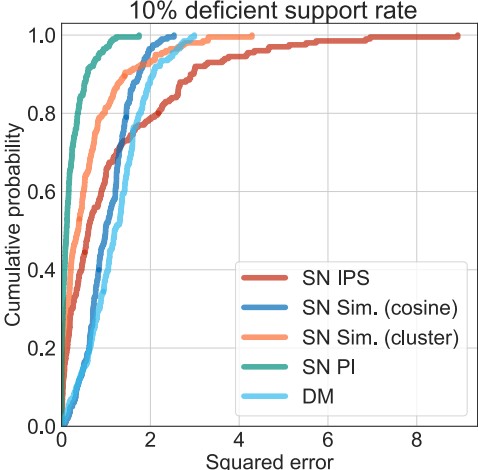

Figure 1: MSE ($\times 10^5$) of the estimators for each deficient support rate.

Figure 2: Cumulative Distribution Functions of the squared error ($\times 10^6$) for 10% deficient support rate.

|                    | 10%  | 20%  | 30%  | 40%  | 50%  | 60%  | 70%  | 80%  | 90%  |
|--------------------|------|------|------|------|------|------|------|------|------|
| SN IPS             | 5.22 | 3.83 | 3.50 | 2.91 | 2.78 | 3.43 | 4.11 | 5.17 | 6.72 |
| DM                 | 3.60 | 2.60 | 2.14 | 1.65 | 1.43 | 1.32 | 1.51 | 1.60 | 1.73 |
| SN PI              | **1.00** | **1.00** | **1.00** | **1.00** | **1.00** | 2.04 | 3.39 | 8.30 | 7.29 |
| SN Sim. (cosine)   | 2.96 | 2.16 | 1.72 | 1.28 | 1.13 | **1.00** | **1.00** | **1.00** | **1.00** |
| SN Sim. (cluster)  | 2.59 | 1.95 | 1.93 | 1.39 | 1.47 | 1.68 | 1.88 | 2.24 | 2.90 |

Table 1: $\text{CVaR}_{0.7}$ with various deficient support rates. Values are normalized for each deficient support rate, the best result is highlighted in bold.

Assumption 4). In this way, the resulting $\bar{\pi}$ satisfies $\sum_{a \in A_0(x)} \bar{\pi}(a|x) = 1$. Hence, the Self-Normalized version of the estimator is consistent under the same assumptions of Theorem 2 [61].

The second proposal is to create a clustering of the actions, which induces a partition of $A$. This is done by applying *K-Means Clustering* [17, 34] (we set $k = 30$) applied on the normalized action feature vectors $f(a)/\|f(a)\|_2$. Let us define the function $c(a)$ that returns, for each action, its corresponding cluster. Then, we can set the weighting function $w_x(a, a') \propto \mathbf{1}(a \in c(a'))$, which is 1 if both $a$ and $a'$ are in the same cluster, and 0 otherwise. Again, we scale the weighting function such that $\sum_{a' \in A_0(x)} w_x(a, a') = 1$. This type of estimator can also be derived starting from an alternative assumption, as we show in Appendix C.

The choice of the similarity function is fundamental for the proposed algorithm. However, implementing a cross-validation procedure for selecting the similarity is not trivial. We design a possible cross-validation procedure in the presence of deficient support, and we present it in Appendix B.1. The idea is based on simulating deficient support on the logged dataset before selecting the similarity function. Notice that, while it represents a promising direction for future research, this proposal has some limitations, which we discuss in Appendix B.1.

## 4.4 Results

We repeat our bootstrap evaluation for various deficient support rates: $\{10\%, 20\%, \ldots, 90\%\}$, with random seeds $\mathcal{S} = \{1, 2, \ldots, 200\}$, for a total of 1800 different combinations. Further details on the infrastructure used and computation time can be found in Appendix B. We compare three estimators (PseudoInverse, Similarity with cosine, Similarity with clustering) with the IPS baseline. We use Self-Normalized variants for each estimator because they outperformed non-normalized

ones. As a regression-based baseline, we include the Direct Method (DM) estimator[3]. We conducted experiments with two different regression models: logistic regression and LightGBM [24]. LightGBM consistently outperformed the logistic regression model; thus, we show the results obtained for DM with LightGBM. We compute the Mean Squared Error (MSE) for each deficient support rate by averaging the squared errors obtained with different seeds[4]. Figure 1 compares the MSE of the evaluated estimators, varying the deficient support rate. From Figure 1, we see how the Similarity estimators dominate the IPS and the DM baselines for each deficient support rate. For a low rate (10%), the estimator with cosine similarity has more or less the same error as the IPS. Nonetheless, increasing the deficient support, the error of IPS increases greatly, and the proposed Similarity estimators are clear winners if compared to IPS. The clustering estimator has a lower MSE with respect to the cosine one for low deficient support rates. For high deficient support rates (>50%), it has a higher error than the cosine one but is still lower with respect to IPS. Also, when we have high support deficiency, we notice that the DM estimator is preferable to the clustering one. This shows how the Similarity estimator with clustering displays a hybrid behavior between the vanilla IPS and the cosine Similarity estimator. This finding suggests that whenever we have deficient support with low rates, the clustering estimator is preferable; conversely, the one with cosine similarity performs better with high deficient support rates. Regarding the PseudoInverse, it has the best MSE up until 50% of deficient support rate. Instead, for high deficient support (higher than or equal to 70%), it displays an unstable behavior with a high variance. This effect may be caused by the violation of the full support on side information assumption, which is shown in Table 2, Appendix B.

For a more interpretable evaluation, we compute the Cumulative Distribution Function (CDF) of the squared errors of the estimators for each deficient support rate, as done by, e.g., [48, 61, 68]. The plot of the computed CDF[5] for a deficient support rate of $10\%$ is in Figure 2. This curve sheds light on a noteworthy aspect of the evaluated estimators, as we explain in the following. Looking solely at Figure 1, we see how the MSE of the Similarity estimator with cosine is almost equal to the value of the standard IPS for a deficient support rate of 10%. The MSE, though, is not a desirable metric when we are in a risk-sensitive scenario [8], i.e., whenever we want to minimize the error in the worst case. From Figure 2 we notice that, while having a similar MSE, our Similarity estimator with cosine is highly preferable to IPS in the worst case (IPS has a much longer tail).

For a better evaluation of the worst case, we show the *Conditional Value-at-Risk$_\alpha$* (CVaR$_\alpha$) metric [1] ($\alpha \in [0, 1)$). This metric computes the average error in the worst $(1 - \alpha) \cdot 100\%$ case. It is widely used in risk-sensitive applications (e.g., in financial portfolio optimization [28, 69]). We set $\alpha = 0.7$ (following [50]), meaning that we evaluate the worst $30\%$ outcomes of the squared error[6]. The results for the various deficient support rates are shown in Table 1. We can see how both Similarity variants outperform the IPS baseline, particularly when we have a strong presence of deficient actions. This indicates that the Similarity estimators are suitable also in risk-sensitive applications with deficient support. The PseudoInverse estimator has an even lower CVaR$_{0.7}$ than Similarity estimators until $50\%$ of deficient support. For higher deficient support rates, it suffers a strong performance degradation. For high deficient support rates. the DM estimator has a lower CVaR$_{0.7}$ than the clustering one, but the best estimator is the Similarity with cosine.

## 5  Related Work

We focus on the mathematical framework of *contextual bandits* [30, 67] with stochastic rewards, which is a specific instance of the more general *Reinforcement Learning* framework [59]. We address the *Off-Policy Evaluation* (OPE) problem, which consists in evaluating a new policy from data logged by a different policy. This problem traces its roots in causal inference [22, 42], and it is related to the estimation of the average treatment effect [19, 20]. Several Off-Policy estimators have been proposed recently for contextual bandit logged data, as this is a very active research area [13, 14, 50, 57, 58, 62, 65, 68]. One of the most used approaches is the Inverse Propensity Score (IPS) estimator [43, 60, 64], which constitutes the heart of many other proposed state-of-the-art estimators [13, 14, 56, 57, 61, 66, 68]. These estimators all rely on the full support assumption, which

---

[3]In Appendix B, we include results also for the *Doubly Robust* (DR) estimator [13], which is an estimator with both a regression-based component and a IPS component.

[4]In Appendix B, we show the MSE obtained with different dataset sizes (50,000 and 150,000).

[5]The CDF plots for all the deficient support rates are in the Appendix B.

[6]Alternative choices of $\alpha$ can be found in Appendix B.

we argued is challenging to obtain in a real-world application. Thus, our paper focuses on OPE with deficient support. Despite its relevance in real-world applications, this setting is not very explored in the research literature.

London and Joachims [35] suggested an approach for improved OPE estimators whenever there is a *"new action"* problem, i.e., when the action set is expanded after the logging phase. This problem is a specialization of the deficient support problem because it focuses on actions that the logging policy could not take for any context. Instead, in the deficient support scenario, the action set with no support may be different for different contexts. Therefore, our approach is more general. Furthermore, they focus only on improving regression-based estimators using theoretical tools from domain adaptation [3, 21], while we concentrate on estimators with no regression model.

Sachdeva et al. [47] are the first to address the problem of deficient support with contextual bandit feedback data. In particular, they propose three techniques for learning in this scenario. Their most successful technique is a learning method based on the IPS estimator. This means that they do not rely on regression models, as done in this paper. However, they focus only on Off-Policy Learning. Hence, their method can not be used for evaluating other policies but only for learning one. Our work, instead, proposes solutions for Off-Policy Evaluation. OPE can be the first step in a learning procedure that maximizes the given estimator.

In the estimators proposed in this paper, we use side information about the actions. Within the scope of bandits, the term *"side information"* is often used with different meanings. In principle, the contextual bandit problem was called *"bandit with side information"* [30, 67] to distinguish it from the standard multi-armed bandit problem. In our paper, we differentiate contexts and side information about the actions: a context consists of information given at each round before selecting an action, while the side information about the actions is known to the decision-maker and fixed for each action. This kind of side information is actively exploited in some *online* bandit algorithms, which make the assumption that there is a metric space on the actions with a Lipschitz distance function of the rewards [25, 26, 41, 54, 55]. Our approach, instead, focuses on the *offline* scenario.

For the concentration analysis, we took inspiration from Metelli et al. [37] and their follow-up papers [38, 39], where they analyzed the theoretical properties of the IPS-like estimators using theoretical tools from information theory, such as the Rényi divergence [44].

## 6   Conclusions

In this paper, we focused on Off-Policy Evaluation, paying special attention to the scenario where the full support assumption is violated. We presented two estimators for a better evaluation by exploiting side information about the actions: one of them was an adaptation of an estimator proposed for a different application domain, while the other is a novel contribution of this paper. We evaluated their theoretical guarantees under alternative assumptions without relying on full support. We also provided finite-sample concentration inequalities, relaxing all the assumptions. The experimental evaluation with real-world data showed that the estimators using side information outperform (both in the average-case and in the worst-case scenario) the traditional approaches (like IPS and regression-based estimators) in the presence of deficient support. For future work, a natural direction to follow is to extend this work to the Off-Policy Learning setting by using the proposed estimators. Furthermore, we presented a novel estimator with hyperparameters to be selected. We proposed a preliminary approach for data-driven hyperparameter selection, but it comes with some limitations. Hence, further research effort in this direction is necessary.

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
