*Proof.* Starting from the similarity assumption, we can rewrite the reward function as follows:

$$
\begin{aligned}
\delta(x, a) &= \sum_{a' \in A_0(x)} w_x(a, a') \delta(x, a') && \text{(similarity assumption)} \\
&= \sum_{a' \in A_0(x)} \frac{\pi_0(a'|x)}{\pi_0(a'|x)} w_x(a, a') \delta(x, a') \\
&= \underset{a' \sim \pi_0(\cdot|x)}{\mathbb{E}} \left[ \frac{w_x(a, a')}{\pi_0(a'|x)} \delta(x, a') \right] && \text{(from the definition of } A_0(x) \text{ and of expected value)}
\end{aligned}
$$

Hence, we can empirically estimate the reward as $\hat{\delta}(x_i, a) := \frac{w_{x_i}(a, a_i)}{\pi_0(a_i|x_i)} r_i$.

Now, recall the definition of the expected reward of the evaluation policy:

$$R(\pi_e) = \underset{x \sim p(\cdot)}{\mathbb{E}} \underset{a \sim \pi_e(\cdot|x)}{\mathbb{E}} [\delta(x, a)] = \underset{x \sim p(\cdot)}{\mathbb{E}} \left[ \sum_{a \in A_e(x)} \pi_e(a|x) \delta(x, a) \right]$$

If we replace the expected value the empirical mean (which is an unbiased estimator of the expected value) we obtain:

$$\hat{R}(\pi_e) = \frac{1}{n} \sum_{i=1}^{n} \sum_{a \in A_e(x_i)} \pi_e(a|x_i) \delta(x_i, a)$$

This is not useful, because $\delta(x_i, a)$ is in general unknown. Nevertheless, we can replace it with its empirical estimator $\hat{\delta}(x_i, a)$ (defined before) and we finally get an unbiased estimator of the expected reward:

$$\hat{R}_S(\pi_e) = \frac{1}{n} \sum_{i=1}^{n} \sum_{a \in A_e(x_i)} \pi_e(a|x_i) \hat{\delta}(x_i, a) = \frac{1}{n} \sum_{i=1}^{n} \sum_{a \in A_e(x_i)} \pi_e(a|x_i) \frac{w_{x_i}(a, a_i)}{\pi_0(a_i|x_i)} r_i$$

$$= \frac{1}{n} \sum_{i=1}^{n} \frac{r_i}{\pi_0(a_i|x_i)} \sum_{a \in A_e(x_i)} \pi_e(a|x_i) w_{x_i}(a, a_i) = \frac{1}{n} \sum_{i=1}^{n} \frac{r_i}{\pi_0(a_i|x_i)} \underset{a \sim \pi_e(\cdot|x_i)}{\mathbb{E}} [w_{x_i}(a, a_i)]$$

$$:= \frac{1}{n} \sum_{i=1}^{n} \frac{\bar{\pi}(a_i|x_i)}{\pi_0(a_i|x_i)} r_i$$

This derivation proves unbiasedness by construction. An alternative proof can be the following:

$$\mathop{\mathbb{E}}_{\substack{x \sim p(\cdot) \\ a \sim \pi_0(\cdot|x) \\ r \sim p(\cdot|x,a)}} \left[ \hat{R}_S(\pi_e) \right] = \mathop{\mathbb{E}}_{\substack{x \sim p(\cdot) \\ a \sim \pi_0(\cdot|x) \\ r \sim p(\cdot|x,a)}} \left[ \frac{\bar{\pi}(a|x)}{\pi_0(a|x)} r \right] \qquad \text{(because data are i.i.d.)}$$

$$= \mathop{\mathbb{E}}_{\substack{x \sim p(\cdot) \\ a \sim \pi_0(\cdot|x)}} \left[ \frac{\bar{\pi}(a|x)}{\pi_0(a|x)} \delta(x,a) \right] \qquad \text{(by definition of } \delta(x,a))$$

$$= \mathop{\mathbb{E}}_{\substack{x \sim p(\cdot) \\ a \sim \pi_0(\cdot|x)}} \left[ \frac{\delta(x,a)}{\pi_0(a|x)} \sum_{a' \in A_e(x)} \pi_e(a'|x) w_x(a',a) \right] \qquad \text{(by definition of } \bar{\pi})$$

$$= \mathop{\mathbb{E}}_{x \sim p(\cdot)} \left[ \sum_{a \in A_0(x)} \delta(x,a) \sum_{a' \in A_e(x)} \pi_e(a'|x) w_x(a',a) \right] \qquad \text{(by definition of expected value)}$$

$$= \mathop{\mathbb{E}}_{x \sim p(\cdot)} \left[ \sum_{a' \in A_e(x)} \pi_e(a'|x) \sum_{a \in A_0(x)} \delta(x,a) w_x(a',a) \right]$$

$$= \mathop{\mathbb{E}}_{x \sim p(\cdot)} \left[ \sum_{a' \in A_e(x)} \pi_e(a'|x) \delta(x,a') \right] \qquad \text{(similarity assumption)}$$

$$= \mathop{\mathbb{E}}_{\substack{x \sim p(\cdot) \\ a' \sim \pi_e(\cdot|x)}} \left[ \delta(x,a') \right]$$

$$= R(\pi_e)$$

$\square$

In the following Lemma, we show that the Support Divergence bounds the estimation bias of the IPS estimator.

**Lemma 1.** *Let us consider a generic off-policy evaluation problem with the expected reward function* $\delta(x,a) \in [0,1]$ *and the standard IPS estimator* $\hat{R}_{IPS}(\pi_e) = \frac{1}{n} \sum_{i=1}^{n} \frac{\pi_e(a_i|x_i)}{\pi_0(a_i|x_i)} r_i$. *Then:*

$$\epsilon^{IPS} := \left| \mathbb{E}\left[ \hat{R}_{IPS}(\pi_e) \right] - R(\pi_e) \right| \le d^{supp}(\pi_e \| \pi_0)$$

*Proof.*

$$\left| \mathbb{E}\left[ \hat{R}_{IPS}(\pi_e) \right] - R(\pi_e) \right| = \left| \mathop{\mathbb{E}}_{\substack{x \sim p(\cdot) \\ a \sim \pi_0(\cdot|x)}} \left[ \frac{\pi_e(a|x)}{\pi_0(a|x)} \delta(x,a) \right] - \mathop{\mathbb{E}}_{\substack{x \sim p(\cdot) \\ a \sim \pi_e(\cdot|x)}} \left[ \delta(x,a) \right] \right|$$

$$= \left| \mathop{\mathbb{E}}_{x \sim p(\cdot)} \left[ - \sum_{a \in \text{Un}(x,\pi_e,\pi_0)} \pi_e(a|x) \delta(x,a) \right] \right| \qquad \text{(from Proposition 1)}$$

$$= \mathop{\mathbb{E}}_{x \sim p(\cdot)} \left[ \sum_{a \in \text{Un}(x,\pi_e,\pi_0)} \pi_e(a|x) \delta(x,a) \right] \qquad \text{(since } \pi_e(a|x)\delta(x,a) \ge 0)$$

$$\le \mathop{\mathbb{E}}_{x \sim p(\cdot)} \left[ \sum_{a \in \text{Un}(x,\pi_e,\pi_0)} \pi_e(a|x) \right] \qquad \text{(since } \delta(x,a) \in [0,1])$$

$$=: d^{supp}(\pi_e \| \pi_0)$$

$\square$

**Proposition 2.** *Let us consider a generic off-policy evaluation problem with side information. Let $\hat{R}_{IPS}$ be the IPS estimator, $\hat{R}_S$ the similarity estimator with weighting function $w_x$, $\hat{R}_{PI}$ the pseudo-inverse estimator. Then, for any $\gamma \in (0,1)$, the following inequalities hold with probability at least $1 - \gamma$:*

$$\left| \hat{R}_{IPS}(\pi_e) - R(\pi_e) \right| \leq d^{supp}(\pi_e || \pi_0) + \sqrt{\frac{d_2(\pi_e || \pi_0)}{n\gamma}},$$

$$\left| \hat{R}_S(\pi_e) - R(\pi_e) \right| \leq \epsilon^S + \sqrt{\frac{d_2(\bar{\pi} || \pi_0)}{n\gamma}},$$

$$\left| \hat{R}_{PI}(\pi_e) - R(\pi_e) \right| \leq \epsilon^{PI} + \sqrt{\frac{2\sigma_{PI}^2 \ln(2/\gamma)}{n}} + \frac{2(\rho_{PI}+1)\ln(2/\gamma)}{3n},$$

*where $\bar{\pi}(a|x) := \mathbb{E}_{a' \sim \pi_e(\cdot|x)}[w_x(a', a)]$, $\sigma_{PI}^2 := \mathbb{E}_{x \sim p(\cdot)}[\mathbf{q}_{\pi_e,x}^T \mathbf{\Gamma}_{\pi_0,x}^\dagger \mathbf{q}_{\pi_e,x}]$, $\rho_{PI} := \sup_x \sup_a |\mathbf{q}_{\pi_e,x}^T \mathbf{\Gamma}_{\pi_0,x}^\dagger f(a)|$, and $\mathbf{q}_{\pi_e,x}, \mathbf{\Gamma}_{\pi_0,x}$ are defined as in Theorem 1.*

*Proof.* We start by proving the first inequality regarding the IPS estimator. First, we bound the variance of $\hat{R}_{\text{IPS}}$ with the exponentiated Rényi divergence, as done in Lemma 4.1 by Metelli et al. [37].

$$
\begin{aligned}
\text{Var}(\hat{R}_{\text{IPS}}(\pi_e)) &= \frac{1}{n} \text{Var}\left( \frac{\pi_e(a|x)}{\pi_0(a|x)} r \right) && \text{(since data is i.i.d.)} \\
&\leq \frac{1}{n} \mathbb{E}\left[ \left( \frac{\pi_e(a|x)}{\pi_0(a|x)} r \right)^2 \right] \\
&\leq \frac{1}{n} \mathbb{E}_{\substack{x \sim p(\cdot) \\ a \sim \pi_0(\cdot|x)}} \left[ \left( \frac{\pi_e(a|x)}{\pi_0(a|x)} \right)^2 \right] && \text{(since } r \in [0,1]) \\
&=: \frac{1}{n} d_2(\pi_e || \pi_0)
\end{aligned}
\tag{1}
$$

Now, we can see that:

$$
\begin{aligned}
\left| \hat{R}_{\text{IPS}}(\pi_e) - R(\pi_e) \right| &\leq \left| \mathbb{E}\left[ \hat{R}_{\text{IPS}}(\pi_e) \right] - R(\pi_e) \right| + \left| \hat{R}_{\text{IPS}}(\pi_e) - \mathbb{E}\left[ \hat{R}_{\text{IPS}}(\pi_e) \right] \right| && \text{(triangle inequality)} \\
&= \epsilon^{\text{IPS}} + \left| \hat{R}_{\text{IPS}}(\pi_e) - \mathbb{E}\left[ \hat{R}_{\text{IPS}}(\pi_e) \right] \right| && \text{(by definition of } \epsilon^{\text{IPS}}) \\
&\leq d^{supp}(\pi_e || \pi_0) + \left| \hat{R}_{\text{IPS}}(\pi_e) - \mathbb{E}\left[ \hat{R}_{\text{IPS}}(\pi_e) \right] \right| && \text{(from Lemma 1)}
\end{aligned}
\tag{2}
$$

Now, for any $\gamma \in (0,1)$, we can apply Chebyshev inequality to the random variable $\hat{R}_{\text{IPS}}(\pi_e)$. We obtain that, with probability at least $1 - \gamma$:

$$
\begin{aligned}
\left| \hat{R}_{\text{IPS}}(\pi_e) - \mathbb{E}\left[ \hat{R}_{\text{IPS}}(\pi_e) \right] \right| &\leq \sqrt{\frac{\text{Var}(\hat{R}_{\text{IPS}}(\pi_e))}{\gamma}} && \text{(Chebyshev)} \\
&\leq \sqrt{\frac{d_2(\pi_e || \pi_0)}{n\gamma}} && \text{(from Eq. 1)}
\end{aligned}
$$

Summing all together with Eq. 2, we obtain the first inequality. The second inequality is obtained with the same procedure. For the third one, it suffices to notice that:

$$\left| \hat{R}_{\text{PI}}(\pi_e) - R(\pi_e) \right| \leq \epsilon^{\text{PI}} + \left| \hat{R}_{\text{PI}}(\pi_e) - \mathbb{E}\left[ \hat{R}_{\text{PI}}(\pi_e) \right] \right|$$

by the triangle inequality, and then apply Theorem 1 of [63] to bound $\left| \hat{R}_{\text{PI}}(\pi_e) - \mathbb{E}\left[ \hat{R}_{\text{PI}}(\pi_e) \right] \right|$.

$\square$

**Definition 1.** *Let $x \in X$ be any fixed context. We say that $w_x$ satisfies the identity of indiscernibles if, for any pair $a_1, a_2 \in A_e(x)$, the following condition is satisfied:*

$$(\forall a' \in A_0(x), \ w_x(a_1, a') = w_x(a_2, a')) \iff a_1 = a_2.$$

**Proposition 3.** *Let us define the reward function $\delta$ restricted to a fixed context $x \in X$ as $\delta_x : A_e(x) \to [0, 1]$, where $\delta_x(a) := \delta(x, a)$. If the similarity assumption holds and $w_x$ satisfies Definition 1, then $\delta_x$ is Lipschitz-continuous (with constant 1) with respect to the metric space $(A_e(x), D)$, where the metric $D$ is defined as $D(a_1, a_2) := \sum_{a' \in A_0(x)} |w_x(a_1, a') - w_x(a_2, a')|$*

*Proof.* First, recall that, because of the similarity assumption, for a given context $x$:

$$\exists w_x : A_e(x) \times A_0(x) \to \mathbb{R} \text{ such that } \forall a \in A_e(x), \ \delta(x, a) = \sum_{a' \in A_0(x)} w_x(a, a')\delta(x, a')$$

Hence, for any $a_1, a_2 \in A_e(x)$:

$$|\delta_x(a_1) - \delta_x(a_2)| := |\delta(x, a_1) - \delta(x, a_2)|$$

$$= \left| \sum_{a' \in A_0(x)} w_x(a_1, a')\delta(x, a') - \sum_{a' \in A_0(x)} w_x(a_2, a')\delta(x, a') \right| \quad \text{(similarity assumption)}$$

$$= \left| \sum_{a' \in A_0(x)} (w_x(a_1, a') - w_x(a_2, a'))\delta(x, a') \right|$$

$$\leq \sum_{a' \in A_0(x)} |(w_x(a_1, a') - w_x(a_2, a'))\delta(x, a')| \quad \text{(triangle inequality)}$$

$$\leq \sum_{a' \in A_0(x)} |w_x(a_1, a') - w_x(a_2, a')| \quad \text{(since } \delta(x, a') \in [0, 1])$$

$$:= D(a_1, a_2)$$

Now, all we have left to do is to note that $D$ is a metric on $A_e(x)$. This is true if $D$ satisfies the following three properties:

- (P1) Identity of indiscernibles: $D(a_1, a_2) = 0 \iff a_1 = a_2$
  - $D(a_1, a_2)$ is a sum of terms $\geq 0$. Hence, the only way for $D(a_1, a_2)$ to be 0 is that $\forall a' \in A_0(x) \ |w_x(a_1, a') - w_x(a_2, a')| = 0$. This is verified if and only if $a_1 = a_2$ because $w_x$ satisfies the identity of indiscernibles (Definition 1) by hypothesis.

- (P2) Symmetry: $D(a_1, a_2) = D(a_2, a_1)$
  - Follows from the symmetry of the difference in absolute value.

- (P3) Triangle inequality: $D(a_1, a_3) \leq D(a_1, a_2) + D(a_2, a_3)$
  - $D(a_1, a_3) := \sum_{a' \in A_0(x)} |w_x(a_1, a') - w_x(a_3, a')| \, \delta(x, a') = \sum_{a' \in A_0(x)} |w_x(a_1, a') - w_x(a_2, a') + w_x(a_2, a') - w_x(a_3, a')| \, \delta(x, a') \leq \sum_{a' \in A_0(x)} (|w_x(a_1, a') - w_x(a_2, a')| + |w_x(a_2, a') - w_x(a_3, a')|)\delta(x, a') = D(a_1, a_2) + D(a_2, a_3)$.

$\square$

**Corollary 1.** *Let us consider the same conditions of Proposition 3 except for the identity of indiscernibles condition on $w_x$. Now, the same $D$ defined in Proposition 3 is a pseudo-metric. Hence, we say that $\delta_x$ is pseudo-Lipschitz-continuous with respect to the pseudo-metric space $(Ae(x), D)$.*

*Proof.* This follows trivially from the proof of Proposition 3, without proving property P1 of the pseudo-metric $D$. $\square$

**Algorithm 1** Deficient Support Evaluation with Real-World Data

---

**Input:** Dataset $\mathcal{D}_r$ logged with a random policy, action set $A$, number of bootstrap samples $n^*$, deficient support rate $p$, an estimator to be evaluated $\hat{R}$, an evaluation policy $\pi_e$, on-policy ground-truth of the policy value $R_{\mathrm{on}}(\pi_e)$, a set of random seeds $\mathcal{S}$
**Output:** A set of squared errors $\mathcal{Z}$
1: $\mathcal{Z} \leftarrow \emptyset$ (initialize set of results)
2: **for** $s \in \mathcal{S}$ **do**
3:     $\mathcal{D}^* \leftarrow \mathrm{Bootstrap}(\mathcal{D}_r; s, n^*)$                      ▷ sample $n^*$ data points
4:     $\mathcal{D}^* \leftarrow \mathrm{DiscardDeficientActions}(\mathcal{D}^*; s, p)$    ▷ discard $p$ percent of actions for each context
5:     $\mathcal{D}^* \leftarrow \mathrm{RebalancePropensity}(\mathcal{D}^*; p)$         ▷ set $\pi_0(a_i|x_i) = 1/|A|(100\% - p)$
6:     $z' \leftarrow \left( R_{\mathrm{on}}(\pi_e) - \hat{R}(\pi_e; \mathcal{D}^*) \right)^2$       ▷ calculate the squared error of the estimator
7:     $\mathcal{Z} \leftarrow \mathcal{Z} \cup \{z'\}$
8: **end for**
9: Return $\mathcal{Z}$

---

| Def. supp. rate | 10% | 20% | 30% | 40% | 50% | 60% | 70% | 80% | 90% |
|---|---|---|---|---|---|---|---|---|---|
| Unsupp. feat. rate | 1.86% | 3.97% | 6.47% | 9.62% | 13.70% | 19.30% | 27.28% | 39.30% | 59.11% |

Table 2: Percentage of unsupported features for varying deficient support rates.

## B   Experiments

In this section, we provide additional notes and results on the experimental evaluation.

**Infrastructure**   We employed an instance called `c6a.8xlarge` from AWS EC2, with 32 cores and 64GB of RAM for our experiments. The Operating System was Ubuntu 20.04. With a dataset size of 100,000 data points, the baselines (SN IPS, Direct Method, and Doubly Robust) and the two Similarity variants (the one with cosine and the other with clustering) were run in parallel; the total computation time for all the 1800 combinations (200 random seeds and 9 deficient support rates) was about 4 hours. For the PseudoInverse, the total computation time for all the 1800 combinations was approximately 12 hours. Furthermore, two sets of additional experiments were run: one with a dataset size $n^* = 50,000$ and the other with a dataset size $n^* = 150,000$. If we sum the computation time of all the experiments, the total computation time is approximately 48 hours.

**Further details on the dataset**   Open Bandit Dataset (OBD) has been released with Open Bandit Pipeline under CC-BY 4.0 license[7]. OBD contains logged bandit feedback from a Japanese large-scale fashion e-commerce platform called ZOZOTOWN[8]. All user features (that compose the context vector) are anonymized with hash functions.

**Further details on the pre-processing phase**   In Algorithm 1, we can see a schematic explanation of the pre-processing procedure we use for the deficient support evaluation. We use $\mathcal{S} = \{1, \ldots, 200\}$, $p \in \{10\%, \ldots, 90\%\}$. For the experiments reported in the main paper, we set $n^* = 100,000$. In the following, we will also show some additional results obtained with $n^* = 50,000$ and $n^* = 150,000$. In Table 2, we show how there can be unsupported features when we increase the deficient support rate ($n^* = 100,000$). We computed the number of unsupported features, for each deficient support rate, averaging the number of features with 0 probability for each seed in $\mathcal{S}$.

**Experimental concentration analysis**   In Table 3, we show the estimated divergence $d_{supp}$, varying the deficient support rate. This value is estimated as:

$$\hat{d}_{supp}(\pi_e || \pi_0) = \frac{1}{n} \sum_{i=1}^{n} \sum_{a \in \mathrm{Un}(x_i, \pi_e, \pi_0)} \pi_e(a|x_i)$$

---

[7]Available at: `https://research.zozo.com/data.html`
[8]`https://zozo.jp/`

| Def. supp. rate | 10% | 20% | 30% | 40% | 50% | 60% | 70% | 80% | 90% |
|---|---|---|---|---|---|---|---|---|---|
| $\hat{d}_{supp}(\pi_e\|\pi_0)$ | 9.73% | 19.72% | 29.88% | 39.72% | 49.68% | 59.81% | 69.87% | 79.89% | 90.00% |

Table 3: Estimated support divergence for different deficient support rates ($n^* = 100,000$).

| Def. supp. rate | 10% | 20% | 30% | 40% | 50% | 60% | 70% | 80% | 90% |
|---|---|---|---|---|---|---|---|---|---|
| $\hat{d}_2(\pi_e\|\pi_0)$ | 6.13 | 6.15 | 6.14 | 6.21 | 6.19 | 6.19 | 6.20 | 6.21 | 6.31 |
| $\hat{d}_2(\bar{\pi}\|\pi_0)$ (cosine) | 1.44 | 1.82 | 2.38 | 3.23 | 4.65 | 7.26 | 12.89 | 28.92 | 114.54 |
| $\hat{d}_2(\bar{\pi}\|\pi_0)$ (cluster) | 4.14 | 5.14 | 6.33 | 7.76 | 9.78 | 12.27 | 15.68 | 20.61 | 27.62 |

Table 4: Estimated exponentiated Rényi divergence for different deficient support rates ($n^* = 100,000$).

We can notice how $\hat{d}_{supp}(\pi_e\|\pi_0)$ is almost equal to the deficient support rate $p$, which is expected: we hide a percentage $p$ of actions recommended by the logging policy, which is a uniform random policy. Therefore, we will hide approximately $p$ percent of data.

In Table 4, we show the estimated divergence $d_2$, varying the deficient support rate, for IPS estimator and the Similarity estimators (both with cosine and clustering). This value is estimated as:

$$\hat{d}_2(\pi\|\pi_0) = \frac{1}{n}\sum_{i=1}^{n}\left(\frac{\pi(a_i|x_i)}{\pi_0(a_i|x_i)}\right)^2$$

where $\pi$ is substituted with $\pi_e$, $\bar{\pi}$ (computed with the cosine), $\bar{\pi}$ (computed with clustering). We can see that the value $\hat{d}_2(\pi_e\|\pi_0)$ does not change too much while varying the deficient support. This effect is expected since the deficient support adds bias to the IPS estimator, and not variance. Instead, we see a raise of $\hat{d}_2(\bar{\pi}\|\pi_0)$ when the deficient support rate increases. This is due to the fact that the Similarity estimators are reducing the high bias of IPS at the cost of some variance.

In Table 5, we show the three estimated values of the bounds given by the concentration inequalities we found in Section 3.3, for $\gamma = 0.05$ (i.e., the inequalities hold with probability $\geq 95\%$). Importantly, we could not estimate the approximation error terms for the similarity estimators $\epsilon^S$ since we have no data-dependent bound for them. Conversely, the IPS approximation error can be bounded from $d_{supp}(\pi_e\|\pi_0)$, which can be easily estimated from data. Therefore, we use the estimated $\hat{d}_{supp}(\pi_e\|\pi_0)$ for the first bound, while the last two bound estimations are optimistic ($\epsilon^S = 0$). Even if it is not fair to compare those three estimated quantities, we can still see how the bias of the IPS takes a dominant role in the computation of its bound.

**Additional experiments** In Figure 3, we plot the MSE obtained while varying the dataset size. In those plots we included also the *Doubly Robust* (DR) baseline. We notice that, with a small dataset ($n^* = 50,000$), the performance of PI degrades faster with the deficient support rate. Also, we notice how DR has intermediate performance between DM and SN IPS, as expected. In Figure 4 and Figure 5, we can see the plots of the CDFs of the estimators for each deficient support rate ($n^* = 100,000$). From each plot, we clearly notice how the CDF of IPS shows, in the worst case, a worse behavior with respect to the two Similarity estimators. This again indicates that, in a risk-averse scenario, we should prefer the Similarity estimators to IPS. Regarding the PseudoInverse, we have an unstable behavior

| Def. supp. rate | 10% | 20% | 30% | 40% | 50% | 60% | 70% | 80% | 90% |
|---|---|---|---|---|---|---|---|---|---|
| $\hat{d}_{supp}(\pi_e\|\pi_0) + \sqrt{\frac{\hat{d}_2(\pi_e\|\pi_0)}{n\gamma}}$ | 0.13 | 0.24 | 0.34 | 0.44 | 0.55 | 0.65 | 0.76 | 0.88 | 1.01 |
| $\sqrt{\frac{\hat{d}_2(\bar{\pi}\|\pi_0)}{n\gamma}}$ (cosine) | 0.02 | 0.02 | 0.03 | 0.03 | 0.04 | 0.06 | 0.09 | 0.17 | 0.48 |
| $\sqrt{\frac{\hat{d}_2(\bar{\pi}\|\pi_0)}{n\gamma}}$ (cluster) | 0.03 | 0.04 | 0.04 | 0.05 | 0.06 | 0.08 | 0.10 | 0.14 | 0.24 |

Table 5: Estimated upper bounds for different deficient support rates, with $\gamma = 0.05$ ($n^* = 100,000$).

for high deficient support rates ($\geq 70\%$) . This suggests that we should avoid the PseudoInverse estimator whenever we have a high deficient support rate.

In Tables 6, 7, 8, and 9, we can see the estimators' CVaR values for $\alpha$ of 0.7, 0.8, 0.9, and 0.95, respectively ($n^* = 100,000$). Let us focus on the $\text{CVaR}_{0.95}$ (Table 9). In this case, we focus on the average squared error in the worst 5% case. This scenario is of substantial interest because it clearly displays how the Similarity estimator is the best one for a risk-sensitive application. Indeed, looking at Figure 1, we see that the MSE of the PseudoInverse is the lowest one for a 40% deficient support rate. However, when we look at the worst 5% of the outcomes, we should prefer the Similarity estimator with cosine, as exhibited in Table 9.

We ran an additional experiment to investigate the behavior of the PI estimator when we pre-processed the provided features. We used random projection [7] with different numbers of components (10, 15, 20, 25, 30), and we compared the performance of PI applied to the pre-processed features with PI applied to the original 40 features with no pre-processing. We focused on the scenario with a support deficiency rate of 50% and repeated the experiments 50 times with different random seeds, $n^* = 100,000$. In Table 10, we report the MSE for each estimator variant. The results show how the original PI has the best performance, but it is comparable with the results obtained with a random projection with 15 or 20 components.

|  | 10% | 20% | 30% | 40% | 50% | 60% | 70% | 80% | 90% |
|---|---|---|---|---|---|---|---|---|---|
| SN IPS | $2.91 \cdot 10^{-6}$ | $3.24 \cdot 10^{-6}$ | $3.82 \cdot 10^{-6}$ | $4.72 \cdot 10^{-6}$ | $5.54 \cdot 10^{-6}$ | $8.28 \cdot 10^{-6}$ | $1.11 \cdot 10^{-5}$ | $1.82 \cdot 10^{-5}$ | $3.76 \cdot 10^{-5}$ |
| DM | $2.01 \cdot 10^{-6}$ | $2.20 \cdot 10^{-6}$ | $2.33 \cdot 10^{-6}$ | $2.67 \cdot 10^{-6}$ | $2.85 \cdot 10^{-6}$ | $3.20 \cdot 10^{-6}$ | $4.09 \cdot 10^{-6}$ | $5.63 \cdot 10^{-6}$ | $9.70 \cdot 10^{-6}$ |
| DR | $2.16 \cdot 10^{-6}$ | $2.34 \cdot 10^{-6}$ | $2.81 \cdot 10^{-6}$ | $3.17 \cdot 10^{-6}$ | $3.35 \cdot 10^{-6}$ | $5.36 \cdot 10^{-6}$ | $7.08 \cdot 10^{-6}$ | $9.54 \cdot 10^{-6}$ | $1.78 \cdot 10^{-5}$ |
| SN PI | $\mathbf{5.57 \cdot 10^{-7}}$ | $\mathbf{8.45 \cdot 10^{-7}}$ | $\mathbf{1.09 \cdot 10^{-6}}$ | $\mathbf{1.62 \cdot 10^{-6}}$ | $\mathbf{1.99 \cdot 10^{-6}}$ | $4.93 \cdot 10^{-6}$ | $9.16 \cdot 10^{-6}$ | $2.92 \cdot 10^{-5}$ | $4.08 \cdot 10^{-5}$ |
| SN Sim. (cosine) | $1.65 \cdot 10^{-6}$ | $1.83 \cdot 10^{-6}$ | $1.87 \cdot 10^{-6}$ | $2.07 \cdot 10^{-6}$ | $2.25 \cdot 10^{-6}$ | $\mathbf{2.42 \cdot 10^{-6}}$ | $\mathbf{2.70 \cdot 10^{-6}}$ | $\mathbf{3.52 \cdot 10^{-6}}$ | $\mathbf{5.60 \cdot 10^{-6}}$ |
| SN Sim. (cluster) | $1.45 \cdot 10^{-6}$ | $1.65 \cdot 10^{-6}$ | $2.11 \cdot 10^{-6}$ | $2.26 \cdot 10^{-6}$ | $2.94 \cdot 10^{-6}$ | $4.05 \cdot 10^{-6}$ | $5.07 \cdot 10^{-6}$ | $7.88 \cdot 10^{-6}$ | $1.63 \cdot 10^{-5}$ |

Table 6: $\text{CVaR}_{0.7}$ ($n^* = 100,000$), the best result is highlighted in bold.

|  | 10% | 20% | 30% | 40% | 50% | 60% | 70% | 80% | 90% |
|---|---|---|---|---|---|---|---|---|---|
| SN IPS | $3.54 \cdot 10^{-6}$ | $4.03 \cdot 10^{-6}$ | $4.65 \cdot 10^{-6}$ | $5.86 \cdot 10^{-6}$ | $6.92 \cdot 10^{-6}$ | $1.04 \cdot 10^{-5}$ | $1.39 \cdot 10^{-5}$ | $2.39 \cdot 10^{-5}$ | $5.17 \cdot 10^{-5}$ |
| DM | $2.19 \cdot 10^{-6}$ | $2.39 \cdot 10^{-6}$ | $2.55 \cdot 10^{-6}$ | $2.94 \cdot 10^{-6}$ | $3.13 \cdot 10^{-6}$ | $3.67 \cdot 10^{-6}$ | $4.67 \cdot 10^{-6}$ | $6.48 \cdot 10^{-6}$ | $1.09 \cdot 10^{-5}$ |
| DR | $2.68 \cdot 10^{-6}$ | $2.94 \cdot 10^{-6}$ | $3.52 \cdot 10^{-6}$ | $3.97 \cdot 10^{-6}$ | $4.26 \cdot 10^{-6}$ | $6.80 \cdot 10^{-6}$ | $8.77 \cdot 10^{-6}$ | $1.19 \cdot 10^{-5}$ | $2.22 \cdot 10^{-5}$ |
| SN PI | $\mathbf{6.86 \cdot 10^{-7}}$ | $\mathbf{1.05 \cdot 10^{-6}}$ | $\mathbf{1.32 \cdot 10^{-6}}$ | $\mathbf{2.04 \cdot 10^{-6}}$ | $\mathbf{2.41 \cdot 10^{-6}}$ | $6.14 \cdot 10^{-6}$ | $1.14 \cdot 10^{-5}$ | $4.05 \cdot 10^{-5}$ | $5.86 \cdot 10^{-5}$ |
| SN Sim. (cosine) | $1.78 \cdot 10^{-6}$ | $2.01 \cdot 10^{-6}$ | $2.04 \cdot 10^{-6}$ | $2.29 \cdot 10^{-6}$ | $2.50 \cdot 10^{-6}$ | $\mathbf{2.75 \cdot 10^{-6}}$ | $\mathbf{3.09 \cdot 10^{-6}}$ | $\mathbf{4.03 \cdot 10^{-6}}$ | $\mathbf{6.73 \cdot 10^{-6}}$ |
| SN Sim. (cluster) | $1.78 \cdot 10^{-6}$ | $2.01 \cdot 10^{-6}$ | $2.54 \cdot 10^{-6}$ | $2.81 \cdot 10^{-6}$ | $3.73 \cdot 10^{-6}$ | $5.18 \cdot 10^{-6}$ | $6.51 \cdot 10^{-6}$ | $1.01 \cdot 10^{-5}$ | $2.15 \cdot 10^{-5}$ |

Table 7: $\text{CVaR}_{0.8}$ ($n^* = 100,000$), the best result is highlighted in bold.

|  | 10% | 20% | 30% | 40% | 50% | 60% | 70% | 80% | 90% |
|---|---|---|---|---|---|---|---|---|---|
| SN IPS | $4.53 \cdot 10^{-6}$ | $5.25 \cdot 10^{-6}$ | $6.23 \cdot 10^{-6}$ | $7.76 \cdot 10^{-6}$ | $9.11 \cdot 10^{-6}$ | $1.45 \cdot 10^{-5}$ | $1.88 \cdot 10^{-5}$ | $3.54 \cdot 10^{-5}$ | $8.56 \cdot 10^{-5}$ |
| DM | $2.48 \cdot 10^{-6}$ | $2.67 \cdot 10^{-6}$ | $2.84 \cdot 10^{-6}$ | $3.31 \cdot 10^{-6}$ | $3.56 \cdot 10^{-6}$ | $4.44 \cdot 10^{-6}$ | $5.46 \cdot 10^{-6}$ | $7.71 \cdot 10^{-6}$ | $1.25 \cdot 10^{-5}$ |
| DR | $3.55 \cdot 10^{-6}$ | $3.89 \cdot 10^{-6}$ | $5.08 \cdot 10^{-6}$ | $5.58 \cdot 10^{-6}$ | $5.95 \cdot 10^{-6}$ | $9.55 \cdot 10^{-6}$ | $1.21 \cdot 10^{-5}$ | $1.67 \cdot 10^{-5}$ | $3.23 \cdot 10^{-5}$ |
| SN PI | $\mathbf{9.06 \cdot 10^{-7}}$ | $\mathbf{1.36 \cdot 10^{-6}}$ | $\mathbf{1.67 \cdot 10^{-6}}$ | $2.66 \cdot 10^{-6}$ | $3.22 \cdot 10^{-6}$ | $8.54 \cdot 10^{-6}$ | $1.57 \cdot 10^{-5}$ | $6.91 \cdot 10^{-5}$ | $1.09 \cdot 10^{-4}$ |
| SN Sim. (cosine) | $1.99 \cdot 10^{-6}$ | $2.23 \cdot 10^{-6}$ | $2.28 \cdot 10^{-6}$ | $\mathbf{2.63 \cdot 10^{-6}}$ | $\mathbf{2.90 \cdot 10^{-6}}$ | $\mathbf{3.34 \cdot 10^{-6}}$ | $\mathbf{3.60 \cdot 10^{-6}}$ | $\mathbf{4.97 \cdot 10^{-6}}$ | $\mathbf{8.48 \cdot 10^{-6}}$ |
| SN Sim. (cluster) | $2.38 \cdot 10^{-6}$ | $2.56 \cdot 10^{-6}$ | $3.33 \cdot 10^{-6}$ | $3.89 \cdot 10^{-6}$ | $5.32 \cdot 10^{-6}$ | $7.20 \cdot 10^{-6}$ | $9.53 \cdot 10^{-6}$ | $1.35 \cdot 10^{-5}$ | $3.23 \cdot 10^{-5}$ |

Table 8: $\text{CVaR}_{0.9}$ ($n^* = 100,000$), the best result is highlighted in bold.

|  | 10% | 20% | 30% | 40% | 50% | 60% | 70% | 80% | 90% |
|---|---|---|---|---|---|---|---|---|---|
| SN IPS | $5.69 \cdot 10^{-6}$ | $6.48 \cdot 10^{-6}$ | $7.97 \cdot 10^{-6}$ | $9.80 \cdot 10^{-6}$ | $1.16 \cdot 10^{-5}$ | $1.91 \cdot 10^{-5}$ | $2.34 \cdot 10^{-5}$ | $4.85 \cdot 10^{-5}$ | $1.26 \cdot 10^{-4}$ |
| DM | $2.73 \cdot 10^{-6}$ | $2.98 \cdot 10^{-6}$ | $3.10 \cdot 10^{-6}$ | $3.63 \cdot 10^{-6}$ | $3.97 \cdot 10^{-6}$ | $5.04 \cdot 10^{-6}$ | $6.11 \cdot 10^{-6}$ | $8.82 \cdot 10^{-6}$ | $1.41 \cdot 10^{-5}$ |
| DR | $4.38 \cdot 10^{-6}$ | $4.96 \cdot 10^{-6}$ | $6.74 \cdot 10^{-6}$ | $7.39 \cdot 10^{-6}$ | $7.54 \cdot 10^{-6}$ | $1.32 \cdot 10^{-5}$ | $1.66 \cdot 10^{-5}$ | $2.21 \cdot 10^{-5}$ | $4.76 \cdot 10^{-5}$ |
| SN PI | $\mathbf{1.11 \cdot 10^{-6}}$ | $\mathbf{1.67 \cdot 10^{-6}}$ | $\mathbf{2.04 \cdot 10^{-6}}$ | $3.34 \cdot 10^{-6}$ | $4.04 \cdot 10^{-6}$ | $1.19 \cdot 10^{-5}$ | $2.10 \cdot 10^{-5}$ | $1.11 \cdot 10^{-4}$ | $2.02 \cdot 10^{-4}$ |
| SN Sim. (cosine) | $2.15 \cdot 10^{-6}$ | $2.41 \cdot 10^{-6}$ | $2.45 \cdot 10^{-6}$ | $\mathbf{2.90 \cdot 10^{-6}}$ | $\mathbf{3.36 \cdot 10^{-6}}$ | $\mathbf{3.93 \cdot 10^{-6}}$ | $\mathbf{4.12 \cdot 10^{-6}}$ | $\mathbf{5.78 \cdot 10^{-6}}$ | $\mathbf{1.03 \cdot 10^{-5}}$ |
| SN Sim. (cluster) | $2.92 \cdot 10^{-6}$ | $3.16 \cdot 10^{-6}$ | $4.39 \cdot 10^{-6}$ | $5.14 \cdot 10^{-6}$ | $6.93 \cdot 10^{-6}$ | $8.83 \cdot 10^{-6}$ | $1.28 \cdot 10^{-5}$ | $1.60 \cdot 10^{-5}$ | $4.37 \cdot 10^{-5}$ |

Table 9: $\text{CVaR}_{0.95}$ ($n^* = 100,000$), the best result is highlighted in bold.

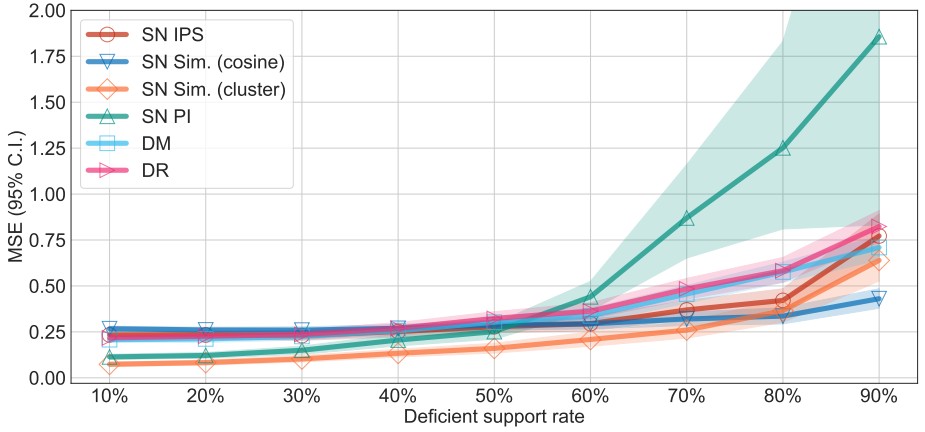

(a) MSE ($\times 10^5$) with dataset size $n^* = 50,000$.

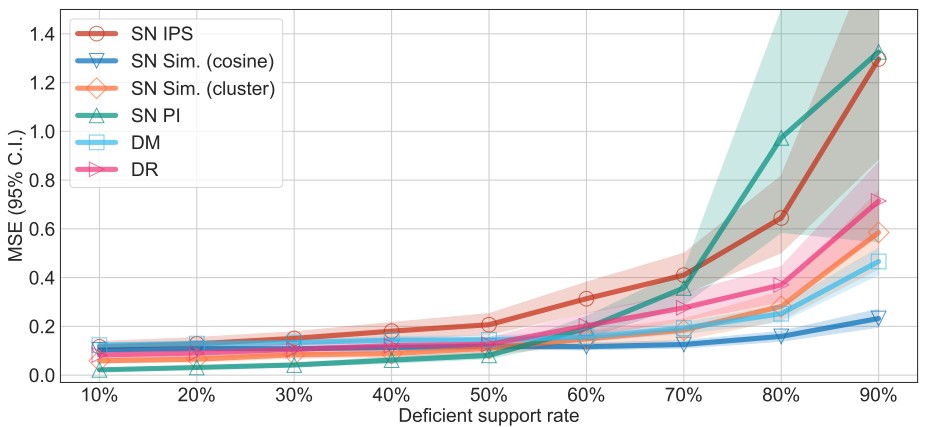

(b) MSE ($\times 10^5$) with dataset size $n^* = 100,000$.

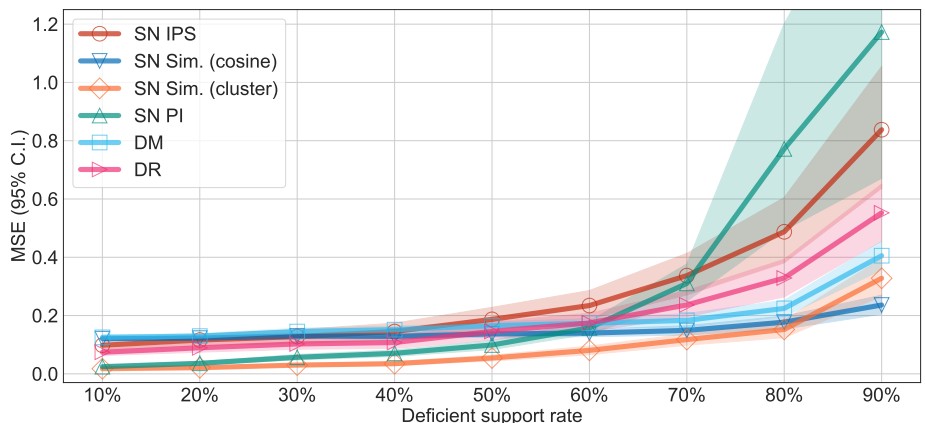

(c) MSE ($\times 10^5$) with dataset size $n^* = 150,000$.

Figure 3: MSE ($\times 10^5$) varying dataset sizes.

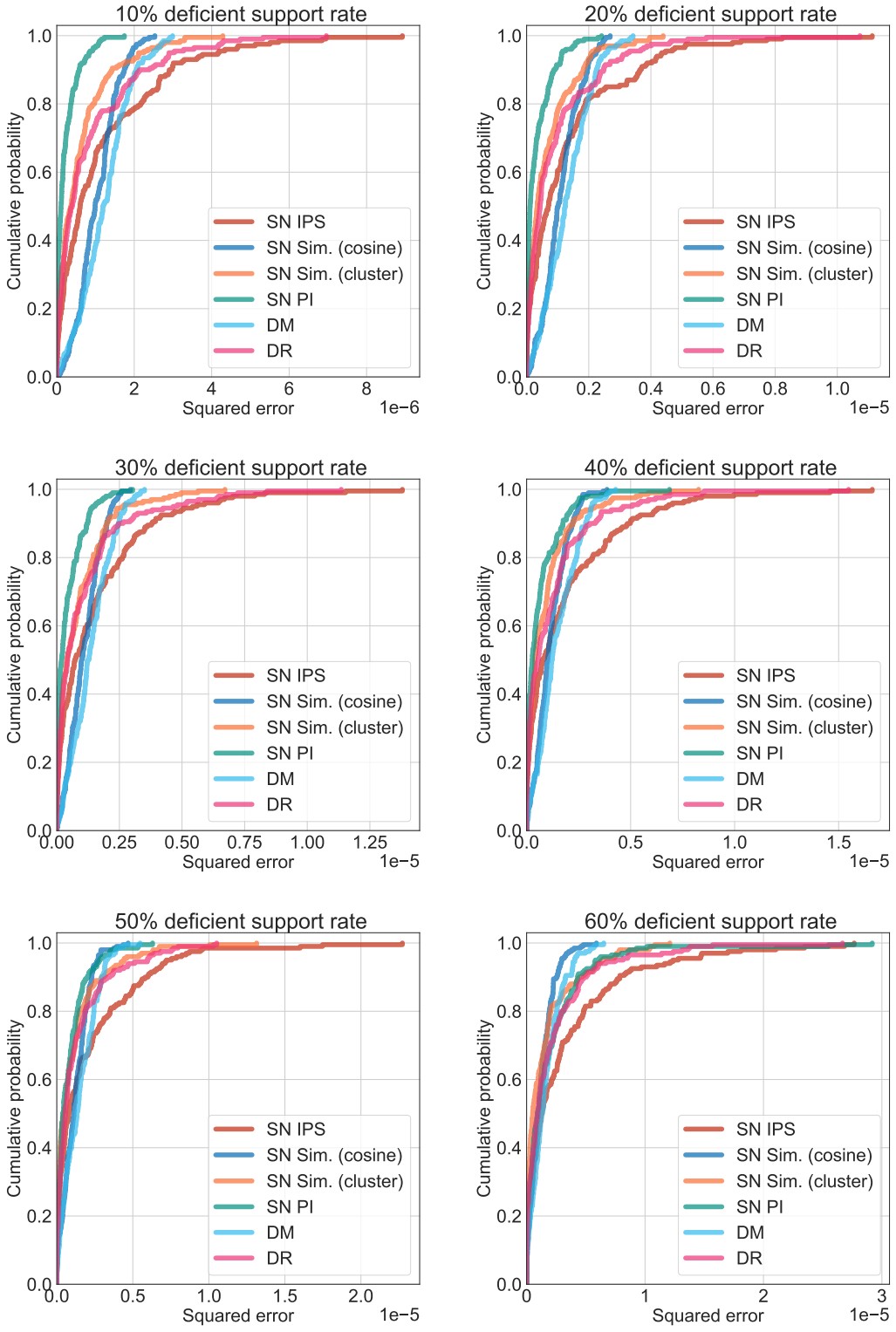

Figure 4: Cumulative Distribution Functions of the squared error varying deficient support rates ($n^* = 100,000$).

| Original | 10 | 15 | 20 | 25 | 30 |
|---|---|---|---|---|---|
| $6.56 \cdot 10^{-7}$ | $1.09 \cdot 10^{-6}$ | $6.91 \cdot 10^{-7}$ | $7.59 \cdot 10^{-7}$ | $1.70 \cdot 10^{-6}$ | $8.72 \cdot 10^{-7}$ |

Table 10: Comparison of the MSEs of PI estimators applied to the original features and to pre-processed features.

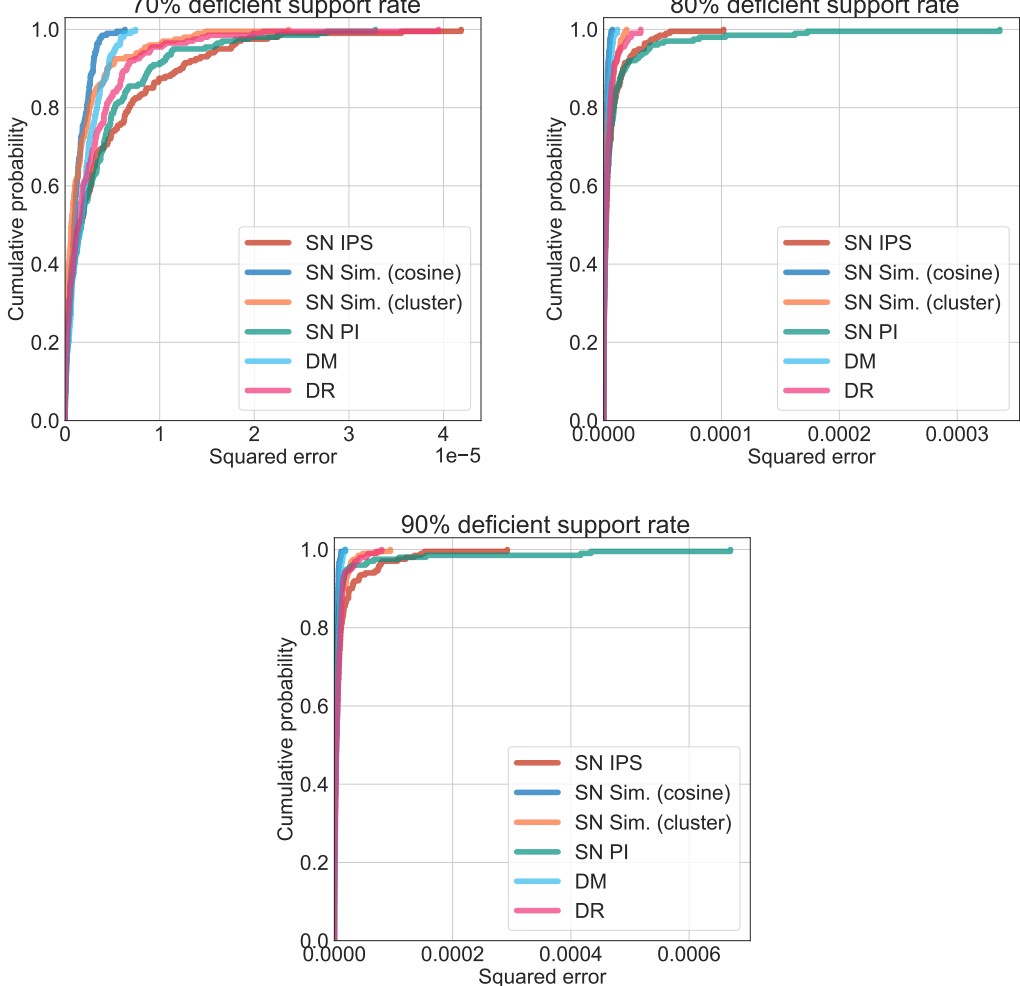

Figure 5: Cumulative Distribution Functions of the squared error varying deficient support rates $(n^* = 100,000)$.

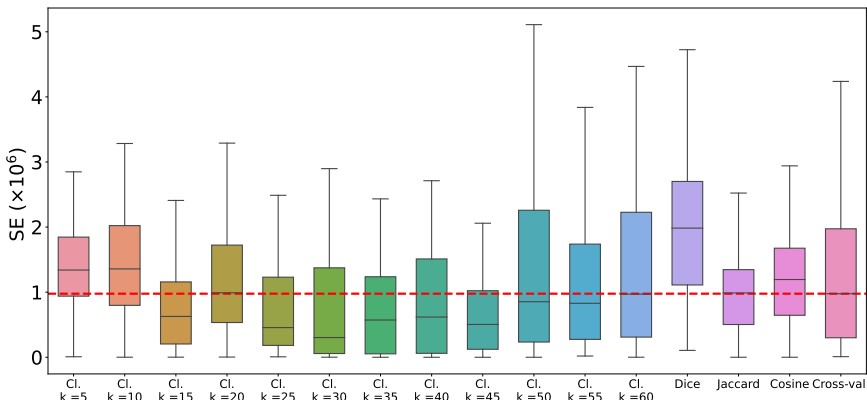

Figure 6: Box plot of the squared errors ($\times 10^6$) obtained by the various similarity estimators with a fixed similarity, compared to the proposed cross-validation procedure. The dashed red line is the median result obtained with the cross-validation procedure.

## B.1 Cross-Validation Proposal

In the following, we design a possible cross-validation procedure in the presence of deficient support.

In the Off-Policy Evaluation problem, we have access to a logged dataset $\mathcal{D} = \{x_i, a_i, \pi_0(a_i|x_i), r_i\}_{i=1}^n$, and we want to evaluate the policy value of a different policy, which we will call *evaluation policy* $\pi_e$. From such a dataset, we can compute an unbiased estimate of the logging policy $\pi_0$ via Monte-Carlo on-policy estimation: $\hat{R}_{on}(\pi_0) = \frac{1}{n} \sum_{i=1}^n r_i$. Let us assume that we know the deficient support rate of $\pi_0$ with respect to $\pi_e$. We call the deficient support rate $p$. We can create another dataset $\mathcal{D}'$ by hiding the $p$ percent of actions from $\mathcal{D}$ for each context $x_i$. In order to simulate the deficient scenario, we should also re-balance the logging policy accordingly. We will call the resulting simulated logging policy $\pi_0'$. Now, we can interpret the original logging policy $\pi_0$ as the evaluation policy (and we know its ground-truth $\hat{R}_{on}(\pi_0)$), and the new $\pi_0'$ as the logging policy. In this way, we can try different variants of the similarity estimator (e.g., varying similarity functions) and evaluate those variants with the error with respect to $\hat{R}_{on}(\pi_0)$. Ultimately, we can choose the best similarity according to which one has the lowest cross-validation error.

**Limitations** We should notice that this approach has two limitations: first, the simulated evaluation policy $\pi_0$ and the simulated logging policy $\pi_0'$ will be very similar. This means that the estimator's validation performance may be optimistic due to low variance. Second, this approach is viable only whenever we have complete knowledge of $\pi_0$ (and not only of the propensities for the logged actions $\pi_0(a_i|x_i)$). Without this knowledge, we can not compute the simulated logging policy $\pi_0'$. To the best of our knowledge, this is the first proposal of a procedure for OPE hyperparameter selection with deficient support. Therefore, it is a research question that needs to address non-trivial issues and requires a broader discussion that could constitute future work.

**Empirical Validation** We ran an additional experiment to validate this procedure empirically. We tried to select the similarity function of the similarity estimator among the following possible choices: Dice, Jaccard, Cosine, and Clustering. For the clustering similarity, we tried different values of $k$ $(5, 10, \ldots, 60)$. We ran this experiment for a dataset with a size $n^* = 100,000$ and 50% of support deficiency. The results are the squared errors obtained over 50 random seeds. We compare the result obtained following the proposed cross-validation procedure with the results obtained by fixing one of the candidate similarities. For each random seed, we may have a different logged dataset. Hence, for each random seed, we repeated the validation procedure 5 times (i.e., we created a simulated logging policy 5 times), and we selected the best similarity according to the best average validation error. Notice how this implies that the cross-validation procedure can possibly choose a different similarity

function for each seed. The squared errors obtained over 50 random seeds are summarized in the box plot in Figure 6.

As expected, the cross-validation procedure displays intermediate results among the possible choices. If we focus on the median result, we see that:

- It performs better than fixing the Dice similarity or the Cosine one. It also outperforms clustering similarities with $k \leq 10$.

- It has comparable performance with the clustering with $k \in \{20, 50, 55, 60\}$ and with the Jaccard similarity.

- It is outperformed by some other similarities, for instance, by the clustering ones with $25 \leq k \leq 45$.

# C   Alternative Derivation for Similarity Estimator with Clustering

In this section, we derive an alternative estimator, which we initially call *Clustering* estimator. In Lemma 2, we show that this is actually a special case of the Similarity estimator, with a particular choice of the weighting function. The derivation is based on the following assumption:

**Assumption 5** (Clustering). *The off-policy evaluation problem satisfies the clustering assumption if, for any context $x$, there exists a partition (called clustering) $C(x) = (c_1, c_2, \ldots, c_k)$ of $A_e(x)$, such that, for any two actions $a, a'$, if they belong to the same cluster ($a, a' \in c$), then, the reward distribution satisfies the following condition: $p(\cdot|x, a) = p(\cdot|x, a') = p(\cdot|x, c)$.*

For simplicity, let us define the cluster membership function as $c(\cdot|x) : A_e(x) \to C(x)$ that returns, for each action, the corresponding cluster. We can view this assumption as the fact that we may have similar actions (which we group into the same cluster) such that the expected reward for those actions is the same: $\delta(x, a_i) = \delta(x, a_j) = \delta(x, c)$ if $c(a_i|x) = c(a_j|x) = c$. In order to derive an unbiased estimator, we also need a full-support assumption on the clustering space.

**Assumption 6** (Full Support on Clustering). *Let us consider an off-policy evaluation problem that satisfies the clustering assumption. The OPE problem satisfies the full support assumption on the clustering if, for any cluster $c \in C(x)$, there is at least one action $a \in c$ that is supported by the logging policy $a \in A_0(x)$, with probability one over $x \sim p(\cdot)$.*

This assumption is less demanding than the standard full support because we only need at least one supported action per cluster.

The following Lemma shows how the Clustering estimator is actually a Similarity estimator with a particular selection of the weighting function.

**Lemma 2.** *Let us consider an off-policy evaluation problem that satisfies the clustering assumption and has full support on clustering. Then, the off-policy problem satisfies also the similarity assumption, with $w_x$ defined as follows:*

$$w_x(a, a') = \frac{\mathbf{1}(a' \in c(a|x))}{\sum_{a_0 \in A_0(x)} \mathbf{1}(a_0 \in c(a|x))}$$

*Proof.* From the clustering assumption, it follows that $\delta(x, a) = \delta(x, c(a|x))$ for any $a$. Hence:

$$
\begin{aligned}
\delta(x, a) &= \delta(x, a) \cdot \frac{\sum_{a_0 \in A_0(x)} \mathbf{1}(a_0 \in c(a|x))}{\sum_{a_0 \in A_0(x)} \mathbf{1}(a_0 \in c(a|x))} && \text{(from the full support on clustering} \\
&&& \text{there is at least one } a_0 \in c(a|x)) \\
&= \frac{\sum_{a_0 \in A_0(x)} \mathbf{1}(a_0 \in c(a|x))\delta(x, a_0)}{\sum_{a_0 \in A_0(x)} \mathbf{1}(a_0 \in c(a|x))} && \text{(because of the clustering assump-} \\
&&& \text{tion, } \delta(x, a_0) = \delta(x, a) \text{ whenever} \\
&&& a_0 \in c(a|x)) \\
&:= \sum_{a_0 \in A_0(x)} w_x(a, a_0)\delta(x, a_0)
\end{aligned}
$$

$\square$

Now, we can present an unbiased estimator for this setting.

**Theorem 3.** *Consider the off-policy evaluation problem where Assumption 5 and Assumption 6 hold. Then, we can define an unbiased estimator $\hat{R}_{CL}(\pi_e)$ of the expected reward of the evaluation policy as:*

$$\hat{R}_{CL}(\pi_e) := \frac{1}{n} \sum_{i=1}^{n} \frac{\pi_e(c_i|x_i)}{\pi_0(a_i|x_i)} \frac{r_i}{N(x_i, c_i)}$$

*where $c_i := c(a_i|x_i)$, $\pi_e(c|x) := \sum_{a \in c} \pi_e(a|x)$, $N(x, c) := \sum_{a' \in A_0(x)} \mathbf{1}(a' \in c)$.*

*Proof.* We start from Lemma 2 and we rewrite the formulation of $\delta(x, a)$ for each $x \in X$ and for each $a \in A_e(x)$.

$$\delta(x, a) = \frac{\sum_{a' \in A_0(x)} \mathbf{1}(a' \in c(a|x))\delta(x, a')}{\sum_{a' \in A_0(x)} \mathbf{1}(a' \in c(a|x))} \qquad \text{(from Lemma 2)}$$

$$= \frac{1}{N(x, c(a|x))} \sum_{a' \in A_0(x)} \mathbf{1}(a' \in c(a|x))\delta(x, a') \qquad \text{(by definition of } N(x, c(a|x)))$$

$$= \frac{1}{N(x, c(a|x))} \sum_{a' \in A_0(x)} \mathbf{1}(a' \in c(a|x))\delta(x, a')\frac{\pi_0(a'|x)}{\pi_0(a'|x)}$$

$$= \frac{1}{N(x, c(a|x))} \mathop{\mathbb{E}}_{a' \sim \pi_0(\cdot|x)} \left[ \frac{\mathbf{1}(a' \in c(a|x))}{\pi_0(a'|x)}\delta(x, a') \right]$$

Now, we can proceed in a similar way as Theorem 2. First, we create an unbiased estimator of the reward by taking a single sample:

$$\hat{\delta}(x_i, a) := \frac{1}{N(x_i, c(a|x))} \frac{\mathbf{1}(a_i \in c(a|x_i))}{\pi_0(a_i|x_i)}r_i$$

Then, we plug $\hat{\delta}(x_i, a)$ inside the empirical mean estimator of the reward of $\pi_e$:

$$\hat{R}_{CL}(\pi_e) = \frac{1}{n} \sum_{i=1}^{n} \sum_{a \in A_e(x_i)} \pi_e(a|x_i)\hat{\delta}(x_i, a)$$

$$= \frac{1}{n} \sum_{i=1}^{n} \sum_{a \in A_e(x_i)} \pi_e(a|x_i)\frac{1}{N(x_i, c(a|x))} \frac{\mathbf{1}(a_i \in c(a|x_i))}{\pi_0(a_i|x_i)}r_i$$

$$= \frac{1}{n} \sum_{i=1}^{n} \frac{r_i}{\pi_0(a_i|x_i)} \sum_{a \in A_e(x_i)} \pi_e(a|x_i)\frac{\mathbf{1}(a_i \in c(a|x_i))}{N(x_i, c(a|x))}$$

Now, we notice that the second summation takes non-zero values only when $a_i \in c(a|x_i)$, i.e., when $c(a|x_i) = c(a_i|x_i) := c_i$. Hence, we can rewrite the sum as:

$$\hat{R}_{CL}(\pi_e) = \frac{1}{n} \sum_{i=1}^{n} \frac{r_i}{\pi_0(a_i|x_i)} \sum_{a \in A_e(x_i)} \pi_e(a|x_i)\frac{\mathbf{1}(a_i \in c(a|x_i))}{N(x_i, c(a|x))}$$

$$= \frac{1}{n} \sum_{i=1}^{n} \frac{r_i}{\pi_0(a_i|x_i)} \sum_{a \in c_i} \frac{\pi_e(a|x_i)}{N(x_i, c_i)}$$

$$= \frac{1}{n} \sum_{i=1}^{n} \frac{r_i}{\pi_0(a_i|x_i)} \frac{1}{N(x_i, c_i)} \sum_{a \in c_i} \pi_e(a|x_i)$$

$$= \frac{1}{n} \sum_{i=1}^{n} \frac{\pi_e(c_i|x_i)}{\pi_0(a_i|x_i)} \frac{r_i}{N(x_i, c_i)} \qquad \text{(by definition of } \pi_e(c_i|x_i))$$

$\square$