# OpenReview forum: "Off-Policy Evaluation with Deficient Support Using Side Information"
_NeurIPS.cc/2022/Conference — NeurIPS 2022 Accept_

### Official Review · Reviewer_yZ68 · 2022-07-10

**Rating:** 5
**Confidence:** 4
**Soundness:** 2 fair
**Presentation:** 3 good
**Contribution:** 2 fair

**Summary:**

This paper studies the off-policy evaluation problem under deficient support, which is a common scenario in industrial settings where action space is large or new actions are added over time. This setting is challenging as the original IS estimator is biased and unreliable. Two estimators are introduced in this setting, which exploit the side information (or action information), and shows different level of bias-variance trade-off. One is based on the pseudo-inverse estimator (PI), which initially designed for the OPE for slates, here basically the idea is to view each feature of the action as a ''slot'', and under the linearity assumption of the reward as well as the coverage on the feature space, the PI gives unbiased estimate. The second estimator is based on action similarities, basically assumes each reward for unsupported actions could be represented by the weighted averages of the rewards of similar actions. Empirical studies based on real-world datasets show better performance (in terms of MSE and CVaR) compared with original self-normalized IPS.

**Questions:**

Please see the Weakness section, and I am willing to adjust my score if they are properly addressed.

**Ethics Review Area:**

["I don’t know"]

**Limitations:**

Yes.

**Strengths And Weaknesses:**

Strength:

1. The paper studies an important problem in the off-policy evaluation literature, i.e., the violation of full-support assumption. Most prior works assume full-support, which is a strong assumption in real-world applications. This paper takes a close look on the problem and discusses new efficient estimators, that based on mild assumptions, such as full-support not in the combinatorial action spaces, but in the feature space, which also shows better empirical performance.

2. I do like the paper's perspective of reducing the combinatorial sense of full support sense in the support over the action features, which draw a nice connection in the slate literature.


Weakness:

1. I am concerned about the applicability of these methods. For PI estimator, it seems highly rely on the discrete features of the action, and the linear assumption of the reward, which seems a reasonable assumption for slate, but seems not the case for the reward for each individual action over features.

2. The similarity estimator seems like the regression-based estimator, which requires extrapolation and generalization (depends on to define the weighting function). It would be great if the authors add more discussion along this line. This is indeed match the experiment result, as we do see better performance of the similarity estimator as the support divergence increases.

3. The only baseline in the empirical results is the self-normalized IPS estimator, in which it is OK if this paper only considers the weighting based approach. However, the proposed similarity estimator shares a lot of similarity with the model-based approach, then it is great if the paper could add more baselines based on the model based approach.

4.  Empirically, it would be also great to add more results in terms of bias, variance to further understand the trade-off of the estimators.

---

> ### Author Response · Authors · 2022-08-02
> **Response to Reviewer yZ68**
>
> We thank the reviewer for the constructive and thoughtful review.
> In the following, we proceed to address the raised concerns.
>
> 1. As the reviewer noticed, the PI estimator is based on the Linearity assumption, which states that the expected reward of each action is a linear combination of the feature of such action, given a fixed context. This assumption justifies a linear regression model for the reward. As in any machine learning problem, relying on the linearity assumption, and thus adopting a linear model, may have pros and cons. The advantages are, for instance, the simplicity of the model, the interpretability, and the computational efficiency. A possible drawback is that not all the phenomena are linear, so a linear model may struggle to model a complex non-linear phenomenon. It is noteworthy that, despite the inherent trade-off, linear models are widely used in the literature. For instance, in the online contextual bandit research field, there are a large number of works that assume linearity of the reward. This setup is often referred to as linear contextual bandits [1].
> Another crucial point is that, although we directly used the side information of the actions in the paper, some (linear or non-linear) transformations may be applied to the side information before computing the PI estimator.
> In this way, such a linear model applied to complex features gains a lot in terms of expressiveness.
> To confirm this point (and also as suggested by Reviewer 4Ega), we ran additional experiments applying Gaussian random projection on the features before computing the PI estimator. We focused on the case with 50\% support deficiency and ran the experiments with 50 different random seeds. The results of this additional experiment can be found in the answer to Reviewer 4Ega, question 4.
>
> [1]: Chu, Wei, et al. "Contextual bandits with linear payoff functions." AISTATS, 2011.
>
> 2. We agree that the similarity estimator has analogies with a regression-based estimator. In the following, we discuss these analogies and also point out some differences among those kinds of estimators. In order to make comparisons among these estimators, we will analyze their expected value. Whenever we omit the distribution of the expectation, we mean that the expectation is w.r.t. the logging data distribution. We define the regression-based estimator, which we call Direct Method (DM), as
> $$ \hat R_{DM}(\pi_e) = \frac{1}{n} \sum_{i=1}^n \sum_{a \in A_e(x_i)} \pi_e(a|x_i) \hat{\delta}(x_i, a) , $$
> where $ \hat{\delta} $ is a regression model. The expected value of DM is
> $$ E[\hat R_{DM}(\pi_e)] = E_x E_{a \sim \pi_e(\cdot|x)} \left[  \hat{\delta}(x, a) \right] $$
> Let us analyze the expected value of the similarity estimator:
> $$ E[\hat R_{S}(\pi_e)]= E_x E_{a' \sim \pi_0(\cdot|x)} \left[ \frac{\bar{\pi}(a'|x)}{\pi_0(a'|x) } \delta(x, a') \right] = E_x \left[ \sum_{a' \in A_0(x)} \bar{\pi}(a'|x) \delta(x, a') \right] = E_x \left[ \sum_{a' \in A_0(x)} \sum_{a \in A_e(x)} \pi_e(a|x) w_x(a, a')  \delta(x, a') \right]  = E_x \left[  \sum_{a \in A_e(x)}  \pi_e(a|x) \sum_{a' \in A_0(x)} w_x(a, a')  \delta(x, a') \right] = E_x E_{a \sim \pi_e(\cdot|x)} \left[  \sum_{a' \in A_0(x)} w_x(a, a')  \delta(x, a') \right]$$
> Therefore, we notice that, in expectation, the similarity estimator is equivalent to a regression-based estimator where the regression model is $\hat{\delta}(x, a) = \sum_{a' \in A_0(x)} w_x(a, a')  \delta(x, a')$. What the similarity estimator is doing (in expectation) is trying to overcome support deficiency by exploiting the expected reward on supported actions and the information on the similarity among supported and unsupported actions. The quality of the generalization of the similarity estimator will depend on the quality of the similarity function. One may think that, because of this analogy, importance sampling may not be required and it suffices to create a regression-based estimator with the regression model $\hat{\delta}(x, a) = \sum_{a' \in A_0(x)} w_x(a, a')  \delta(x, a')$. The problem is that it is not possible to directly estimate all the possible $\delta(x, a')$ from data without importance sampling.
> For further discussion on this, we refer to the answer to question 3 of Reviewer 4ega. We will include the whole discussion on the analogies and differences between these two estimators in the final version of the paper.

---

> > ### Author Response · Authors · 2022-08-02
> > **Response to Reviewer yZ68 (cont.)**
> >
> > 3. As suggested, we ran additional experiments to include regression-based estimators. In particular, we added the standard Direct Method (DM) and the Doubly Robust (DR) estimator. We conducted experiments with two different regression models: logistic regression and LightGBM. LightGBM consistently outperformed the logistic regression model; thus, we show the results obtained with LightGBM.
> > In the following table, we show the results in terms of MSE (the errors are averaged over 200 random seeds) for the various deficient support rates. The dataset is the same used in the original experiments, with a size of 100,000 data points. For each deficient support rate, we underline the estimators with an error lower than the baselines that do not exploit side information (SN IPS, DM, DR), and we highlight in bold the best performing estimator.
> >    $$
> > \begin{array}{|l|ccccccccc|}
> > {} &  10\\% & 20\\% &   30\\% &  40\\% &   50\\% &  60\\% &                        70\\% &                        80\\% &                        90\\% \\\\
> > \hline
> > \text{SN IPS} &  1.16 \cdot 10^{-6} &                    1.28\cdot 10^{-6} &                    1.51\cdot 10^{-6} &                    1.80\cdot 10^{-6} & 2.07\cdot 10^{-6} &                    3.14\cdot 10^{-6} &                    4.10\cdot 10^{-6} &                    6.45\cdot 10^{-6} &     1.30\cdot 10^{-5} \\\\
> > \text{DM} &  1.23\cdot 10^{-6} & 1.29\cdot 10^{-6} &1.34\cdot 10^{-6} &                    1.43\cdot 10^{-6} & 1.45\cdot 10^{-6} &    1.56\cdot 10^{-6} & 1.93\cdot 10^{-6} &   2.52\cdot 10^{-6} &   4.66\cdot 10^{-6} \\\\
> > \text{DR}  &  8.28\cdot 10^{-7} &   9.04\cdot 10^{-7} &  1.06\cdot 10^{-6} & 1.21\cdot 10^{-6} & 1.26\cdot 10^{-6} &  2.03\cdot 10^{-6} &    2.76\cdot 10^{-6} &   3.70\cdot 10^{-6} &  7.14\cdot 10^{-6} \\\\ \text{SN PI}   &  \pmb{\underline{2.16\cdot 10^{-7}}} &  \pmb{\underline{3.14\cdot 10^{-7}}} &  \pmb{\underline{4.15\cdot 10^{-7}}} &  \pmb{\underline{6.15\cdot 10^{-7}}} &  \pmb{\underline{8.10\cdot 10^{-7}}} &                    1.92\cdot 10^{-6} &  3.58\cdot 10^{-6} &  9.73\cdot 10^{-6} &     1.33\cdot 10^{-5} \\\\  \text{SN Sim. (cosine)}  & 1.04\cdot 10^{-6} &  1.09\cdot 10^{-6} &   1.08\cdot 10^{-6} & \underline{1.13\cdot 10^{-6}} & \underline{1.19\cdot 10^{-6}} &  \pmb{\underline{1.17\cdot 10^{-6}}} &  \pmb{\underline{1.26\cdot 10^{-6}}} &  \pmb{\underline{1.59\cdot 10^{-6}}} &  \pmb{\underline{2.32\cdot 10^{-6}}} \\\\ \text{SN Sim. (cluster)} & \underline{5.89\cdot 10^{-7}} &\underline{6.56\cdot 10^{-7}} &\underline{8.37\cdot 10^{-7}} & \underline{8.85\cdot 10^{-7}} &  \underline{1.10\cdot 10^{-6}} & \underline{1.49\cdot 10^{-6}} & \underline{1.85\cdot 10^{-6}} & 2.82\cdot 10^{-6} & 5.86\cdot 10^{-6} \\\\ \hline \end{array} $$
> > This result empirically highlights the analogies and differences between the similarity estimators and the regression-based ones. Like the similarity estimator, the error of DM does not increase much when we increase the deficient support, in particular when support deficiency is low. For instance, by increasing the deficient support rate from 10\% to 60\%, we see an increase of the error of only $0.3 \cdot 10^{-6}$. Interestingly, though, for high deficient support, the regression-based models struggle. For example, by increasing the deficient support rate from 70\% to 90\%, we see an increase in the error of DM of $2.73 \cdot 10^{-6}$. This finding suggests that support deficiency makes a regression-based model inaccurate, particularly with high deficient support. The DR estimator shows a hybrid behavior between the DM and IPS, which is expected. The problem is that IPS is biased when we have deficient support, so DR is worse than plain DM for high deficient support ($\geq$ 60\%).
> > As suggested by Reviewer 4Ega, we replicated this experiment for different dataset sizes (50,000 and 150,000), and the obtained results further confirm these findings.
> >
> > 4. We agree with this observation and we want to point out that a discussion along this line is already present in Appendix B, under the paragraph "Experimental concentration analysis". In the main text of the paper (Section 3.3), we derived three different theoretical concentration bounds. All these bounds share the same structure: there is a component that bounds the bias and another component that bounds the square root of the variance. In the mentioned paragraph in Appendix B, we estimated those bounds empirically, showing how the problem of IPS, in case of deficient support, is biased. At the same time, we can see from those results how the similarity estimators try to reduce the bias at the cost of a slightly higher bias. In Table 3 (Appendix B) there is a comprehensive comparison of those bounds with a probability of at least 95%.
> >
> > We hope that the reviewer is satisfied with our response and clarifications, which addressed all of the raised concerns. We hope the reviewer may consider raising their rating if they are satisfied with our response.

---

> > > ### Comment · Reviewer_yZ68 · 2022-08-09
> > > **Thanks for the author response**
> > >
> > > Thanks for the author response and discussions about the question! I appreciate the additional experiments on DM based method and more refined analysis. It addresses most of my concerns and I am willing to update my score.

---

### Official Review · Reviewer_nksT · 2022-07-10

**Rating:** 6
**Confidence:** 4
**Soundness:** 3 good
**Presentation:** 3 good
**Contribution:** 3 good

**Summary:**

This paper addresses the problem of performing off-policy evaluation using inverse propensity weighting when there are zero probability actions with respect to the logged policy. To address this, the authors provide two alternative assumptions that can be employed to find valid off-policy estimates: positivity under side information and linearity with respect to action features. After employing these, the authors show that previously proposed estimators for off policy evaluation in slate recommendation can be used for off policy evaluation. An additional nearest neighbor style estimator is also provided.

**Questions:**

* Do the authors have thoughts on how a practitioner could test for violations to the assumptions proposed here?
* How does the proposed estimators compare to an outcome model? The similarity model, in particular seems nearly identical to an outcome model. It would be helpful if the authors could include some discussion on this.

**Limitations:**

Limitations are well addressed by the authors.

**Strengths And Weaknesses:**

Strengths:

* This is a very challenging and relevant problem. Positivity violations can occur frequently in practice and there is currently no good options for dealing with this issue in the weighting context.
* Simplicity: the authors clearly lay out the challenge and the solution in a way that makes plain what is required to gain identifiability.
* Strong empirical performance.

Weaknesses:

* The main weakness here is that both assumptions are essentially untestable.
* There are a number of hyper-parameters, it is unclear how practitioners should set these in practice.

---

> ### Author Response · Authors · 2022-08-02
> **Response to Reviewer nksT**
>
> We thank the reviewer for the insightful observations. We respond to the raised concerns below.
>
> - "Do the authors have thoughts on how a practitioner could test for violations to the assumptions proposed here?"
>
>     A violation of the assumptions would lead to a biased estimator and consequently to a higher evaluation error. In standard supervised learning, one of the most used ways of estimating the prediction error is to use cross-validation: hide a part of the training data into a validation set, train the model on the remaining training set, and evaluate the performance on the validation set. In an Off-Policy Evaluation problem, we are not able to perform cross-validation in this way because we do not have access to the ground truth policy value of the evaluation policy. Furthermore, we are in the presence of deficient support. Hence, we proposed a possible solution for a validation procedure with deficient support. The idea is based on simulating deficient support on the logged dataset before selecting the similarity function. The detailed proposal is presented in the general comment. Notice that, while it represents a promising direction for future research, this proposal has few limitations, such as requiring complete knowledge of the logging policy (and not only the propensity of the logged actions).
>
> - "How does the proposed estimators compare to an outcome model? The similarity model, in particular seems nearly identical to an outcome model. It would be helpful if the authors could include some discussion on this."
>
>     We agree with the reviewer that a discussion on the comparison between a regression-based estimator and our similarity estimator would be helpful. Reviewer yZ68 shared the same interest in this matter. Hence, we decided to write a comprehensive discussion on the analogies and differences between the two types of estimators. This discussion can be found in the answer to question 2 by Reviewer yZ68. Furthermore, in response to question 3 by Reviewer yZ68, we also conducted additional experiments to include regression-based baselines, showing that, empirically, the regression-based estimator has a similar behavior compared to our similarity estimator with low deficient support, but it suffers from performance degradation with higher deficient support rates. We will include the complete discussion in the final version of the paper.

---

### Official Review · Reviewer_4Ega · 2022-07-11

**Rating:** 6
**Confidence:** 4
**Soundness:** 3 good
**Presentation:** 3 good
**Contribution:** 2 fair

**Summary:**

The paper looks at the problem of off-policy evaluation where the subset-support assumption is violated. Roughly speaking, the idea to get around the support issue is by estimating rewards for the actions (that may never be seen under the logging policy) using the rewards of the logged actions. For instance, prior methods assume rewards to be parametric (linear) function of action features which allows estimating rewards for the unseen actions using ordinary linear regression. In contrast, the proposed work makes an assumption that allows inferring rewards for the unseen action using a non-parametric/kernel regression. When the assumptions hold exactly, these estimators are unbiased. Empirically, the proposed estimator seems to perform better than the baseline methods when the support is very deficient.


**Questions:**

1\. Line 39: The verbose description of “full support” makes it read like the logging policy needs a non-zero chance of taking every action.  I would recommend making it clear that the logging policy needs a non-zero chance of taking every action _that is possible under the evaluation policy_.


2\.Line 157: Does $w$ not require any constraints at all? Quick eyeballing suggests that it might be required that $w>=0$ and that $\sum_{a}w(a,a’)  = 1$ for all $a’$. Without this it seems like $\bar \pi$ in Theorem 2 may not result in a valid probability distribution and subsequently Renyi divergence for it may become ill-defined in Proposition 2?

3\. Similarity estimator: Is importance sampling necessary? Given that an estimator of rewards can be constructed from Assumption 4, is it not possible to explicitly compute the expectation under the evaluation policy $\pi$?

$\frac{1}{n}\sum_{i=1}^n \sum_a \pi(a|x_i) \delta(x_i, a)$

I can see one argument about minimizing compute time, but I guess sample efficiency is more important for OPE than compute. Further,

 (a) Given that the utility of this method is for large action sets, $pi_0(a|s)$ in the denominator is likely going to have some very small values that may blow up the variance of the estimator relative to explicitly computing the expectation,
(b) Looks like $w$ in the experiments do not depend on $x$. Further, the action set is discrete. These can make explicit computation of the expectation easier?
(c) One could investigate various ways of reducing computation time for directly estimating the expectation, for e.g., use normalized w values to develop sampling based procedure for estimating $\delta(x,a)$

4\. Paper would be stronger if more ablations with different choice of feature vectors (random-projection, cosine basis, etc.) are presented to help the baseline. Doing linear regression on one-hot encoding feels naive.

5\. Paper would also benefit from having ablations on the impact of dataset size on the estimators.

6\. Choosing similarity metric $w$ (or even the underlying feature) appropriately is important for the proposed algorithm to work well. Given that reward function estimation can be reduced to supervised-learning, it would be nice to have a cross-validation procedure for both the baseline and the proposed method implemented while discussing the usefuleness of the results.

7\. Section 4.3: Looks like $w_x$ does not depend on $x$ at all? That does not seem like a good choice in general.

8\. Not sure if I understood the normalization process used in Table 1.


**Limitations:**

Discussed above.

**Strengths And Weaknesses:**

Strength

- The problem may be of interest to a wider audience.
- Natural idea with a clear presentation.

Weakness

- There may be immediate ways to improve the proposed estimator under the assumptions being made.
- Paper could benefit from more thorough empirical investigation

---

> ### Author Response · Authors · 2022-08-02
> **Response to Reviewer 4Ega**
>
> We would like to thank the reviewer for taking the time to write this comprehensive and engaging review. We address the questions in the following:
>
> 1. We agree with the reviewer and we will clarify the description of "full support" in the introduction.
>
> 2. We agree with the reviewer. In order for $\bar{\pi}$ to be a valid probability distribution, $w_x$ should be greater than 0 and should sum up to one. We will update Assumption 4 accordingly.
>
> 3. We thank the reviewer for this interesting question.
>     The reviewer questions if importance sampling is actually necessary given Assumption 4, and proposes to estimate the policy value exploiting Assumption 4 with an explicit computation of the expectation under the evaluation policy. However, we argue in the following that this proposal is not viable in general.
>
> Let us reformulate the proposal of the reviewer.
>     An empirical estimation of the policy value of $\pi_e$ is:
>     $$ \frac{1}{n} \sum_{i=1}^n \sum_{a \in A_e(x_i)} \pi_e(a|x_i) \delta(x_i, a) \  .$$
>     This formulation, with an explicit computation of the expectation under $\pi_e$, requires the knowledge of $\delta(x_i, a)$ for each $x_i$ in the dataset and each $a \in A_e(x_i)$. Of course, this quantity is not known, but Assumption 4 says that, for any pair $x \in X$ , $a \in A_e(x)$, we have $\delta(x, a) = \sum_{a' \in A_0(x)} w_x(a, a') \delta(x, a')$. Therefore, the proposal is to plug this information inside the estimator (replacing $\delta(x_i, a)$):
>     $$ \frac{1}{n} \sum_{i=1}^n \sum_{a \in A_e(x_i)} \pi_e(a|x_i) \sum_{a' \in A_0(x_i)} w_{x_i}(a, a') \delta(x_i, a') $$
> In this way, we do not have the problem of computing $\delta(x_i, a)$ for any $x_i$, $a\in A_e(x_i)$, but we have another problem: the computation/estimation of $\delta(x_i, a')$ for any $x_i$, and any $a' \in A_0(x_i)$. It may seem that this quantity can be estimated from data because now we are focusing on actions supported by the logging distribution. However, the fact that $\pi_0(a'|x_i) > 0$ does not imply that the logging policy actually ever picked $a'$ in the logging phase. Thus, we can not estimate, in general, $\delta(x_i, a')$ from data, because we may have never seen the pair $x_i, a'$, and consequently we may have never observed any $r \sim p(\cdot|x_i, a')$.
>
> Questions regarding analogies and differences between the similarity estimator and a model-based estimator were asked by other reviewers (Reviewer nksT and Reviewer yZ68). We will add a complete discussion on this matter in the final version of the paper.
>
> Regarding the related questions:
>
> a. It is true that for large action sets $\pi_0$ may be small and the variance of an IS-based estimator may be large. However, an IS-based estimator allows many principled techniques to reduce the variance, e.g., with Self-Normalization (while remaining consistent), therefore this issue can be mitigated.
>
> b. Actually, $w$ in the experiments does depend on $x$, but we will expand on this when we answer question 7. Regarding the second part of this question, we argued that the estimator based on the explicit computation of the expectation is not viable.
>
> c. Regarding this point, while we argued that the estimator proposed by the reviewer is not viable, this proposal is still applicable to our similarity estimator. Indeed, our estimator requires an explicit computation of an expectation under $\pi_e$: $R_S(\pi_e) = \frac{1}{n} \sum_i \frac{r_i}{\pi_0(x_i|a_i)} E_{a' \sim \pi_e(\cdot|x_i)} \left[ w_{x_i} (a', a_i) \right]$. Therefore, as pointed out here, we can mention that we can estimate the quantity $E_{a' \sim \pi_e(\cdot|x_i)} \left[ w_{x_i} (a', a_i) \right]$ instead of computing explicitly the expectation. This can be done by sampling $\bar{n}$ actions according to $\pi_e(\cdot|x_i)$. This procedure will maintain the unbiasedness of the estimator, and it can reduce the computation time for large action sets. The number $\bar{n}$ is a parameter that will allow practitioners to regulate the trade-off between evaluation quality and computation time. We will include this discussion in the paper, as we think it may be useful for large action sets.\
>
> 4.  We thank the reviewer for the suggestion. We ran an additional experiment to investigate the behavior of the PI estimator when we pre-process the provided features. We used random projection with different numbers of components (10, 15, 20, 25, 30), and we compared the performance of PI applied to the pre-processed features with PI applied to the original 40 features with no pre-processing. We focused on the scenario with a support deficiency rate of 50\%, and we repeated 50 times the experiments with different random seeds. The dataset used is the one used in the original experiments, with 100,000 data points. In the table below, we report the MSE for each estimator variant.

---

> > ### Author Response · Authors · 2022-08-02
> > **Response to Reviewer 4Ega (cont.)**
> >
> > $$ \begin{array}{|c|ccccc|} \hline \text{Original} &  10 &       15 &       20 &       25 &       30 \\\\ \hline 6.56\cdot 10^{-7} & 1.09\cdot 10^{-6} & 6.91\cdot 10^{-7} & 7.59\cdot 10^{-7} & 1.70\cdot 10^{-6} & 8.72\cdot 10^{-7} \\\\  \hline \end{array}$$ The results show how the original PI has the best performance, but it is comparable with the results obtained with a random projection with 15 or 20 components. Therefore, in this case pre-processing the features does not improve the result.
> >
> > 5. We replicated the experiments done in the paper for different dataset sizes (50,000 and 150,000 data points). Here, we show additional experiments with 150,000 data points.
> > $$\begin{array}{|l|ccccccccc|} \hline {} & 10\\% &20\\% &30\\% &40\\% &50\\% &60\\% &                        70\\% &                        80\\% &90\\% \\\\ \hline \text{SN IPS}&  9.79\cdot 10^{-7} &1.16\cdot 10^{-6} &1.28\cdot 10^{-6} & 1.45\cdot 10^{-6} &1.87\cdot 10^{-6} & 2.34\cdot 10^{-6} &3.37\cdot 10^{-6} & 4.87\cdot 10^{-6} &  8.38\cdot 10^{-6} \\\\
> > \text{DM} & 1.25\cdot 10^{-6} &1.29\cdot 10^{-6} &  1.45\cdot 10^{-6} &1.52\cdot 10^{-6} &1.65\cdot 10^{-6} &                    1.74\cdot 10^{-6} &1.83\cdot 10^{-6} & 2.24\cdot 10^{-6} &4.05\cdot 10^{-6} \\\\
> > \text{DR}& 7.41\cdot 10^{-7} & 8.93\cdot 10^{-7} & 1.03\cdot 10^{-6} &1.07\cdot 10^{-6} &                    1.46\cdot 10^{-6} &1.75\cdot 10^{-6} & 2.36\cdot 10^{-6} &3.29\cdot 10^{-6} & 5.52\cdot 10^{-6} \\\\
> > \text{SN PI}             &        \underline{2.45\cdot 10^{-7}} &        \underline{3.61\cdot 10^{-7}} &        \underline{5.67\cdot 10^{-7}} &        \underline{7.04\cdot 10^{-7}} &        \underline{9.89\cdot 10^{-7}} &        \underline{1.58\cdot 10^{-6}} & 3.10\cdot 10^{-6} &   7.72\cdot 10^{-6} &                    1.17\cdot 10^{-5} \\\\
> > \text{SN Sim. (cosine)}  &                    1.21\cdot 10^{-6} &                    1.22\cdot 10^{-6} &                    1.30\cdot 10^{-6} &                    1.29\cdot 10^{-6} &        \underline{1.37\cdot 10^{-6}} &        \underline{1.40\cdot 10^{-6}} &        \underline{1.49\cdot 10^{-6}} &        \underline{1.77\cdot 10^{-6}} &  \pmb{\underline{2.36\cdot 10^{-6}}} \\\\
> > \text{SN Sim. (cluster)} &  \pmb{\underline{1.76\cdot 10^{-7}}} &  \pmb{\underline{2.10\cdot 10^{-7}}} &  \pmb{\underline{2.99\cdot 10^{-7}}} &  \pmb{\underline{3.48\cdot 10^{-7}}} &  \pmb{\underline{5.43\cdot 10^{-7}}} &  \pmb{\underline{8.05\cdot 10^{-7}}} &  \pmb{\underline{1.17\cdot 10^{-6}}} &  \pmb{\underline{1.52\cdot 10^{-6}}} &        \underline{3.28\cdot 10^{-6}} \\\\
> > \hline
> > \end{array}
> > $$
> > In the tables, for each deficient support rate, we underlined the estimators with a lower error compared to all the baselines that do not explicitly use side information (SN IPS, DM, DR). For each deficient support rate, the lowest MSE is highlighted in bold.
> > From these results, we can see how the observations made for the dataset of size 100,000 still hold. We notice that the estimators actively exploiting side information consistently outperform the baselines with no side information usage. The similarity estimator with clustering has a lower MSE compared to the cosine one for low deficient support rates. Instead, for extremely deficient support, the cosine similarity is preferable. This behavior is in line with the original results. The PI estimator has a behavior similar to what we saw in the original dataset too: for low deficient support rates, it shows strong performance, but for high deficient support rates it suffers from substantial performance degradation.
> > 6. We agree with the fact that the choice of the similarity is fundamental for the proposed algorithm. Nevertheless, implementing a cross-validation for our estimator is not trivial. Such an estimator, as we argued in the answer for question 3, can not be reduced to supervised learning. Therefore, we design a possible cross-validation procedure in the presence of deficient support. The proposal can be found in the general comment, because other reviewers shared the same interest.
> >
> > 7.  To be exact, $w_x$ depends on $x$ because of the normalization procedure: $\sum_{a \in A_0(x)} w_x(a', a) = 1$ for any $a \in A_e(x)$. This means that, since the supported actions for different contexts may be different ($A_0(x) \neq A_0(x')$ in general), $w_x(a', a) \neq w_{x'}(a', a)$. Nevertheless, we agree with the reviewer that our proposal (computing the similarity based solely on the action features, and then normalize) may be suboptimal. However, such similarity is very fast to compute and reached good performance in the experiments. We believe that our formulation is general enough to allow both simple and complex similarity functions and this is an advantage of our proposed estimator.
> >
> > 8. In Table 1, every column is normalized independently based on the best performing estimator for the given deficient support rate. The unnormalized version of Table 1 can be found in Table 6 in Appendix B.

---

> > > ### Comment · Reviewer_4Ega · 2022-08-09
> > > **Thank you for clarifications**
> > >
> > > Thank you for your clarification. Adding these discussions and results in the main paper can be beneficial for the readers. I have updated my score accordingly.

---

### Official Review · Reviewer_p8Mu · 2022-07-12

**Rating:** 6
**Confidence:** 3
**Soundness:** 4 excellent
**Presentation:** 4 excellent
**Contribution:** 3 good

**Summary:**

This paper proposes two algorithms for off-policy evaluation in the presence of actions with no support in the observational data. The first algorithm is adapted from [1] and assumes a parametric decomposition of the expected reward, whereas the second algorithm leverages an action similarity assumption. The authors provide finite sample error guarantees for the two methods (as well as for the state of the art) and  a comprehensive empirical evaluation.

[1] Ray Jiang, Sven Gowal, Yuqiu Qian, Timothy A. Mann, and Danilo J. Rezende. Beyond greedy ranking: Slate optimization via list-cvae.

**Questions:**

Taken from the above section:

* Why are the action features only binary? What changes in the implementation/ theoretical results if the features are continuous?
* Can the weights matrix for the Similarity algorithm be learned rather than assumed?

**Limitations:**

The authors have adequately addressed and potential negative societal impact of their work.

**Strengths And Weaknesses:**

**Strengths**
* This is a well written paper in a not well studied area of OPE
* The presentation is clear and concise
* The experimental study is compelling

**Weaknesses**
* The study is limited to binary features only, with no clear explanation as to why
* The structure of the weights matrix in the Similarity algorithm is assumed (e.g. cosine similarity) rather than learned . Given that the linearity in reward (Assumption 4) is pretty strong, is it possible to instead learn the weighting scheme?
* There is no guidance on which of the two methods to use (depending on the scenario). These methods display different behaviors in the experiments section.

---

> ### Author Response · Authors · 2022-08-02
> **Response to Reviewer p8Mu**
>
> We thank the reviewer for their interesting comments. We address the questions in the following:
>
> - "Why are the action features only binary? What changes in the implementation/ theoretical results if the features are continuous?"
>
> From a theoretical point of view, nothing changes by including continuous features in the theoretical setting. Hence, we thank the reviewer for pointing this out, and we will update the paper to clarify this.
> In the experimental setup, we decided to include only binary features since the datasets contained 40 of them, while only one feature was continuous. Mixing binary and continuous features is indeed possible with cosine, however, the different nature of the feature values may require the introduction of an appropriate normalization to produce good results in practice, hence we decided to avoid this in order to focus on the main goal of the paper.
>
> - "Can the weights matrix for the Similarity algorithm be learned rather than assumed?"
>
> Yes, the weights of the similarity matrix can be learned, but it is not trivial in an Off-Policy scenario.
> A possible way to learn from data the similarity to use could be a cross-validation procedure. We could try different similarity functions and choose the one with the lowest validation error. The problem is that, in an Off-Policy Evaluation problem, we do not have access to the ground truth policy value of the evaluation policy. As a possible solution for a validation procedure, we propose to simulate support deficiency on the logged dataset to select the best performing similarity. This procedure is presented in the general comment in further detail. Notice that, while it represents a promising direction for future research, this proposal has some limitations, such as requiring complete knowledge of the logging policy (and not only the propensity of the logged actions).

---

> > ### Comment · Reviewer_p8Mu · 2022-08-08
> > **Rebuttal Response**
> >
> > Thank you for the response. I stand by my original assessment and will be keeping my score unchanged.

---

### Author Response · Authors · 2022-08-02
**Cross-Validation proposal**

In the following we design a possible cross-validation procedure in the presence of deficient support.

In the Off-Policy Evaluation problem, we have access to a logged dataset $ D = (x_i, a_i, (a_i|x_i), r_i)_{i=1}^n $ and we want to evaluate the policy value of an evaluation policy $\pi_e$. Trivially, we can compute the on-policy estimation for the policy value of $ \pi_0 $: $R(\pi_0) =\frac{1}{n} \sum r_i $. Let us assume that we know the deficient support rate of $\pi_0$ with respect to $\pi_e$. We call the deficient support rate $p$.

We can create another dataset $\mathcal{D}'$ by hiding $p$ percent of actions from $D$ for each context $x_i$. In order to simulate the deficient scenario, we should also re-balance the simulated logging policy $\pi_0'$. Now, we can interpret the original logging policy $\pi_0$ as the evaluation policy (and we know its ground-truth $R(\pi_0)$, and the new $\pi_0'$ as the logging policy. In this way, we can try different variants of the similarity estimator (e.g., with different similarity functions), and evaluate those variants with the error with respect to $R(\pi_0)$.
In the end, we can choose the best similarity according to which one has the lowest cross-validation error.

We should notice that this approach has two limitations: first, the simulated evaluation policy $\pi_0$ and the simulated logging policy $\pi_0'$ will be very similar. This means that the estimator's validation performance may be optimistic, due to low variance. Second, this approach is viable only whenever we have a complete knowledge of $\pi_0$. Without this knowledge, we can not compute the simulated logging policy $\pi_0'$. To the best of our knowledge, this is the first proposal of a procedure for OPE hyperparameter selection with deficient support, therefore it is a research question that requires to address non trivial issues and requires a wider discussion that could constitute a future work.

We ran an additional experiment in order to validate empirically this procedure. We tried to select the similarity function of the similarity estimator among the following possible choices: Dice, Jaccard, Cosine, Clustering. For the clustering similarity, we tried different values of k (5, 10, ..., 60). We ran this experiment for a dataset with size 100,000 and 50\% of support deficiency. First, we show the results obtained by fixing the various candidate similarities. The results are the mean squared errors, obtained by averaging over 50 random seeds.
$$\begin{array}{|ccccccccccccccc|}
    \hline
      \text{Dice} &  \text{Jaccard} &  \text{Cosine} &  \text{Cluster, k=5} & \text{Cluster, k=10} &  \text{Cluster, k=15} &  \text{Cluster, k=20} &  \text{Cluster, k=25} & \text{Cluster, k=30} &  \text{Cluster, k=35} &  \text{Cluster, k=40} &  \text{Cluster, k=45} &  \text{Cluster, k=50} &  \text{Cluster, k=55} &  \text{Cluster, k=60}  \\\\
    \hline
   1.97\cdot 10^{-6} &           1.04\cdot 10^{-6} &            1.23\cdot 10^{-6}  &      1.47\cdot 10^{-6} &              1.52\cdot 10^{-6} &              8.91\cdot 10^{-7} &              1.30\cdot 10^{-6} &              8.37\cdot 10^{-7} &              9.99\cdot 10^{-7} &              1.20\cdot 10^{-6} &              1.26\cdot 10^{-6} &              8.58\cdot 10^{-7} &              1.88\cdot 10^{-6} &              1.65\cdot 10^{-6} &              2.04\cdot 10^{-6}          \\\\   \hline \end{array} $$

Now, we show the results of our cross-validation procedure. For each random seed, we have a different logged dataset, we repeated the validation procedure 5 times, and we selected the best similarity according to the best average validation error. Notice how this implies that the cross-validation procedure can possibly select a different similarity function for each seed.
$$  \begin{array}{|c|}
    \hline
    \text{Cross-Validation}   \\\\
    \hline
                       1.56\cdot 10^{-6}     \\\\
    \hline
    \end{array} $$

As expected, the cross-validation procedure shows an intermediate results among the possible choices: it performs better than fixing the Dice similarity or the clustering with k$\geq$50, it has comparable performance with the clustering with $k \in \{5, 10\}$, and it is outperformed by some other similarities like the Jaccard one.
We will include this validation procedure proposal (along with the experimental results) in the final version of the appendix, since we think it may be valuable for researchers and practitioners, and it is a promising direction for future research.

---

### Meta-Review · Area_Chair_A8Fe · 2022-08-26

**Recommendation:** Accept
**Confidence:** Less certain

**Metareview:**

The reviewers all appreciate the direction of this work, and while the merits and significance of the work have limitations -- as reflected in the weak but positive scores -- all reviewers were positive and found no reason to reject. I agree with this and recommend acceptance on the basis that the merits of the contribution and having this as part of the program outweigh the concerns regarding significance. That said, I strongly encourage the authors to use the detailed feedback from the reviewers to improve their paper.

**Award:**

No

---

### Decision · Program_Chairs · 2022-09-14

Accept